# Causes and Consequences of A Glutamine Induced Normoxic HIF1 Activity for the Tumor Metabolism

**DOI:** 10.3390/ijms20194742

**Published:** 2019-09-24

**Authors:** Matthias Kappler, Ulrike Pabst, Claus Weinholdt, Helge Taubert, Swetlana Rot, Tom Kaune, Johanna Kotrba, Martin Porsch, Antje Güttler, Matthias Bache, Knut Krohn, Fabian Bull, Anne Riemann, Claudia Wickenhauser, Barbara Seliger, Johannes Schubert, Bilal Al-Nawas, Oliver Thews, Ivo Grosse, Dirk Vordermark, Alexander W. Eckert

**Affiliations:** 1Department of Oral and Maxillofacial Plastic Surgery, Martin Luther University Halle-Wittenberg, 06120 Halle (Saale), Germany; pabst.ulrike@gmail.com (U.P.); swetlana.rot@medizin.uni-halle.de (S.R.); tom.kaune@uk-halle.de (T.K.); johanna.kotrba@med.ovgu.de (J.K.); mkg.sekretariat@uk-halle.de (J.S.); al-nawas@uni-mainz.de (B.A.-N.); alexander.eckert@uk-halle.de (A.W.E.); 2Institute of Computer Science, Martin Luther University Halle-Wittenberg, 06120 Halle (Saale), Germany; claus.weinholdt@informatik.uni-halle.de (C.W.); grosse@informatik.uni-halle.de (I.G.); 3Clinic of Urology and Pediatric Urology, FA University Hospital Erlangen-Nürnberg, 91054 Erlangen, Germany; Helge.Taubert@uk-erlangen.de; 4Core Facility Deep Sequencing, Institute of Computer Science, and Institute of Human Genetics, Martin Luther University Halle-Wittenberg, 06120 Halle (Saale), Germany; martin.porsch@informatik.uni-halle.de (M.P.); fabian.bull@informatik.uni-halle.de (F.B.); 5Department of Radiotherapy, Martin Luther University Halle-Wittenberg, 06120 Halle (Saale), Germany; antje.guettler@uk-halle.de (A.G.); matthias.bache@uk-halle.de (M.B.); dirk.vordermark@uk-halle.de (D.V.); 6Core Unit DNA Technologies, University of Leipzig, 04103 Leipzig, Germany; krok@med.uni-leipzig.de; 7Institute of Physiology, Martin Luther University Halle-Wittenberg, 06110 Halle (Saale), Germany; anne.riemann@medizin.uni-halle.de (A.R.); oliver.thews@medizin.uni-halle.de (O.T.); 8Institute of Pathology, Martin-Luther-University Halle-Wittenberg, 06110 Halle (Saale), Germany; claudia.wickenhauser@uk-halle.de; 9Institute of Medical Immunology, Martin-Luther-University Halle-Wittenberg, 06110 Halle (Saale), Germany; barbara.seliger@uk-halle.de; 10German Centre for Integrative Biodiversity Research (iDiv) Halle-Jena-Leipzig, 04103 Leipzig, Germany

**Keywords:** HIF1α, hypoxia, normoxia, glutamine, glutaminolysis, glycolysis, Warburg effect, metabolism, tumor, L-ascorbic acid, acetylsalicylic acid

## Abstract

The transcription factor hypoxia-inducible factor 1 (HIF1) is the crucial regulator of genes that are involved in metabolism under hypoxic conditions, but information regarding the transcriptional activity of HIF1 in normoxic metabolism is limited. Different tumor cells were treated under normoxic and hypoxic conditions with various drugs that affect cellular metabolism. HIF1α was silenced by siRNA in normoxic/hypoxic tumor cells, before RNA sequencing and bioinformatics analyses were performed while using the breast cancer cell line MDA-MB-231 as a model. Differentially expressed genes were further analyzed and validated by qPCR, while the activity of the metabolites was determined by enzyme assays. Under normoxic conditions, HIF1 activity was significantly increased by (i) glutamine metabolism, which was associated with the release of ammonium, and it was decreased by (ii) acetylation via acetyl CoA synthetase (ACSS2) or ATP citrate lyase (ACLY), respectively, and (iii) the presence of L-ascorbic acid, citrate, or acetyl-CoA. Interestingly, acetylsalicylic acid, ibuprofen, L-ascorbic acid, and citrate each significantly destabilized HIF1α only under normoxia. The results from the deep sequence analyses indicated that, in HIF1-siRNA silenced MDA-MB-231 cells, 231 genes under normoxia and 1384 genes under hypoxia were transcriptionally significant deregulated in a HIF1-dependent manner. Focusing on glycolysis genes, it was confirmed that HIF1 significantly regulated six normoxic and 16 hypoxic glycolysis-associated gene transcripts. However, the results from the targeted metabolome analyses revealed that HIF1 activity affected neither the consumption of glucose nor the release of ammonium or lactate; however, it significantly inhibited the release of the amino acid alanine. This study comprehensively investigated, for the first time, how normoxic HIF1 is stabilized, and it analyzed the possible function of normoxic HIF1 in the transcriptome and metabolic processes of tumor cells in a breast cancer cell model. Furthermore, these data imply that HIF1 compensates for the metabolic outcomes of glutaminolysis and, subsequently, the Warburg effect might be a direct consequence of the altered amino acid metabolism in tumor cells.

## 1. Introduction

Aerobic glycolysis (also known as the Warburg effect) enables tumor cells to consume high levels of glucose for proliferation, producing lactate as a waste production, even in the presence of oxygen [1,2]. Thus, aerobic glycolysis bypasses the Krebs cycle (TCA cycle) and it avoids oxidative phosphorylation [3,4] in a metabolic adaptation of tumor cells that enables them to incorporate nutrients into the biomass [4,5].

However, it was suggested that glucose catabolism (aerobic glycolysis) alone is not sufficient for the rapid proliferation of cancer cells. Indeed, the combination of glucose as a carbon source and the amino acid glutamine as a nitrogen source has been shown to accelerate the growth of different tumor types [6,7,8,9]. In particular, tumor cells consume glutamine for the anabolic processes in the synthesis of other amino acids and nucleotides and membranes [7]. In healthy cells, glutamine acts as a carrier of excess nitrogen (e.g., toxic ammonia), which is released from cells into the blood for further processing and excretion via the liver and kidney [6,10,11]. Recently, an association between nitrogen and glucose metabolism during increased glycolysis was postulated as a consequence of nitrogen metabolism [12]. Therefore, glycolysis and glutaminolysis might be linked to metabolic pathways [13]. The transcription factor HIF1 (hypoxia-inducible factor 1) is stabilized and activated under hypoxic conditions. Under hypoxia, the HIF1 α-subunit accumulates and then dimerizes with the HIF1-β-subunit (ARNT) to form the transcription factor HIF1 [14,15,16]. HIF1 is associated with hypoxic metabolism, particularly during increased glycolysis [17,18,19,20]. Under these circumstances HIF1-mediated transcriptional upregulation of almost all glycolytic enzymes under hypoxic conditions (http://www.ncbi.nlm.nih.gov/biosystems/138045?Sel=geneid:8553#show=genes, last accessed on: 17 September 2019). Under normoxia, the α-subunit of HIF1 is not detectable due to its hydroxylation and degradation, which leads to inactivated HIF1. Although various studies have described the accumulation of HIF1 under normoxia [18,21], thus far the function of stabilized normoxic HIF1, including its role in metabolic processes, has not been analyzed in detail [3,16,21,22,23,24,25,26]. Interestingly, Parks and coauthors suggested a possible function of HIF1 in the metabolic adaptation of aerobic glycolysis [27], a finding that is in line with the association of HIF1-mediated transcriptional upregulation of glycolytic genes in a tumor model of Drosophila [25]; in this tumor model, the glycolytic genes were upregulated, not only by the HIF1 associated with hypoxia, but also due to the activity of HIF under normoxia, which seems to be responsible for the development of glycolytic tumors [25]. This assumption is further strengthened by a recent review summarizing the role of HIF1 in creating the Warburg effect [28], which is a normoxic phenomenon. 

Kwon and Lee [29] have already discussed the influence of glutamine and/or glucose on the accumulation of HIF1α under hypoxia. In addition, a connection between the amino acid glutamine, HIF1, and aerobic glycolysis has already been assumed [30]. Our recent studies suggested a link between glutaminolysis and glucose metabolism that could be affected by normoxic HIF1 activity [13,31,32]. We have already described the accumulation and activity of normoxic HIF1 caused by the catabolization of the amino acid glutamine and by the release of the waste product of glutaminolysis, ammonia [13,31,32]. 

The present study compares the effect of glutamine application on tumor cells under normoxic and hypoxic conditions, which leads to the normoxic activity of HIF1 and the involvement of HIF1 in the regulation of glycolysis-associated genes and on metabolites under normoxia and hypoxia. These results indicate that HIF1 has a more general role under normoxic and hypoxic conditions.

## 2. Results

### 2.1. Expression of HIF1α and CAIX under Normoxic and Hypoxic Conditions

Glutaminolysis, the catabolism of the amino acid glutamine, leads to the release of the waste products ammonia/ammonium. Intracellular pH (pH_l_) and its regulation are of central importance in the disposal of these waste products (Appendix A). In previous studies, we demonstrated that ammonia/ammonium are responsible for the normoxic stabilization and activation of HIF1 [13] and that HIF1 might have a function in the detoxification of ammonia/ammonia. Moreover, the role of HIF1 as an intracellular pH sensor has been postulated [31,32].

The normoxic stabilization of HIF1α at the protein level was found in various tumor cell lines after the application of several growth factors (e.g., EGF or insulin) or fetal bovine serum only in the presence of the amino acid glutamine and as a result of glutaminolysis [13]. 

Referring to these findings [13], we validated the effects of glutamine under normoxia on physiological HIF1 activity and aimed to elucidate the underlying molecular mechanisms for these empirical findings. Furthermore, significant genes that were regulated by HIF1 under normoxia and/or hypoxia were identified. For this purpose, the influence of the different components, such as fetal bovine serum (FBS) as growth factor replacement, glutamine, and HIF1α status, as determined by HIF1α silencing via siRNA, under normoxia or hypoxia was analyzed. As shown in Figure 1, the HIF1 target carbonic anhydrase (*CAIX*), a regulator of extracellular pH value, was only detectable in the presence of HIF1α. While glutamine had no effect on the stability of the involved proteins (HIF1 or CAIX) under hypoxic conditions, under normoxia, HIF1 and CAIX were significantly upregulation after the application of glutamine and FBS (Appendix A), while treatment with glutamine alone led to a limited level of expression (Figure 1). We concluded that the expression of HIF1 or CAIX is not hypoxia-specific and that an oxygen-independent mechanism must be responsible for HIF1 stabilization under normoxia.

### 2.2. Role of Acetylation for the Normoxic and Hypoxic Stabilization of HIF1α 

In contrast to the hypoxic situation, the availability of glutamine is responsible for the stabilization of HIF1α under normoxia [13] in different tumor cell lines (Figure 2).

In addition, the stability of HIF1α is also decreased by the acetylation of the α-subunit of HIF1 [33]. Therefore, acetylating agents, such as acetylsalicylic acid or ibuprofen, were used to affect the level of HIF1α in different tumor cells. Both of the added substances significantly reduced the normoxic HIF1α level, which had been induced by applied glutamine (Figure 3, Appendix A). Under hypoxia, this observed effect was not found (Appendix A), but higher levels of these drugs reduced the level of hypoxic-stabilized HIF1 in the MCF-7 cells (Appendix A). While using the very strong acetylating agent phenylacetate (acetic acid phenyl ester (CH_3_COOC_6_H_5_)), the level of HIF1α under hypoxia almost completely disappeared in the cells of different lineages (Appendix A).

The level of cytosolic acetyl-CoA (Ac-CoA) is essential for HIF1α acetylation [33]. The enzymes ATP citrate lyase (ACLY) and the cytosolic form of acetyl-CoA synthetase (ACSS2) generate the cytoplasmic form of the substrate Ac-CoA. Therefore, downregulating ACLY or ACSS2 is expected to affect the available level of cytosolic Ac-CoA and cause glutamine-independent normoxic HIF1α accumulation. 

Indeed, a glutamine-independent normoxic accumulation of HIF1α was found in the SAOS-2 cells after ACLY was knocked down (as compared to the level of accumulation of HIF1α after ACSS2 was downregulated and that of the control) (Figure 4). In contrast, the knocking down of ACSS2 resulted in a substantial glutamine-independent normoxic accumulation of HIF1α in the XF354 and MDA-MB-231 cells when compared to that of the ACLY downregulated cells and the control cells, respectively (Figure 4). 

Three different cell lines were treated with sodium citrate since the level of cytosolic Ac-CoA is usually determined by the mitochondrial citrate level. As shown in Figure 5 and Appendix A, sodium citrate significantly reduced the level of glutamine-induced normoxic HIF1 α-subunit. The application of Ac-CoA in MDA-MB-231 cells resulted in a non-significant reduction of glutamine-induced normoxic HIF1α-level (*p* = 0.13) (Appendix A). 

These data demonstrated that acetylating agents could prevent the glutamine-induced accumulation of HIF1α under normoxia. Vice versa, HIF1 can be stabilized independently of glutamine by influencing the Ac-CoA balance within the cells. In addition, the acetylation of HIF1α might play a role in its stability under hypoxia.

### 2.3. Effect of Ascorbic Acid on the Normoxic and Hypoxic Stabilization of HIF1α

Another mechanism that affects HIF1 stabilization is the hydroxylation of proline residues, which is mediated by proteins of the hypoxia-inducible prolyl hydroxylase (EGLNs) family. EGLNs require oxygen and ascorbic acid. 

Thus, MDA-MB-231 cells were treated with 5 mM glutamine (Figure 6) and with ascorbic acid or HIF1α siRNAs under normoxia and hypoxia. Under normoxia, the ascorbic acid treatment completely reversed (*p* = 0.01) the glutamine-induced HIF1α protein accumulation in a manner that is similar to that of HIF1α siRNA silencing (*p* = 0.03) (Figure 6). Under hypoxia, ascorbic acid did not reduce the HIF1α level in a significant manner (*p* = 0.44), while the treatment with siRNA (*p* < 0.001) under hypoxia significantly reduced the HIF1α protein level, as expected (Figure 6, Appendix A). 

Carbonic anhydrase (*CAIX*) is a transcriptional target of the transcription factor HIF1 [16]. The results from the analysis of *CA9* mRNA levels under normoxia confirms the data regarding HIF1α levels (Figure 6), showing a 71% reduction (*p* = 0.008) due to the application of ascorbic acid (Figure 7). The reduction in the *CA9* mRNA levels was 87% (*p* = 0.003) after the cells were treated with a combination of ascorbic acid and HIF1α siRNA. Under hypoxic conditions, the use of ascorbic acid had no significant effects on the *CA9* mRNA levels (Figure 7). These data show that ascorbic acid affects the accumulation and activity of normoxic HIF1.

### 2.4. Deep Sequencing Analysis of HIF1α-Positive or HIF1α-Negative Cells under Normoxic or Hypoxic Conditions 

The transcript levels of the MDA-MB-231 cells treated with glutamine and/or fetal bovine serum and/or cells transfected with siRNA against HIF1α were determined under normoxic or hypoxic conditions using RNA-seq to identify the role of normoxic HIF1. The results from the analysis of the mRNA samples corresponded to that of the protein samples (four independent experiments), as shown in lanes 7 and 8 for the normoxic conditions and 7* and 8* for the hypoxic conditions (Figure 1).

The mRNA levels of 231 (132+99) transcripts of encoding genes under normoxia and 1384 (1252+132) transcripts of encoding genes under hypoxia were significantly altered, as determined by a log2-fold change ≥1 (Appendix A) when HIF1α was degraded by siRNA treatment when compared to the level of control cells. Another 2246 transcripts of encoding genes were deregulated under hypoxia independently of HIF1 (Appendix A). The KEGG pathway analysis showed that, in particular, glycolysis-associated genes and genes that are involved in carbon metabolism were regulated by HIF1 under hypoxia and normoxia (Appendix A). Of those involved in the glycolytic pathway, 16 of the 1384 differentially expressed hypoxic genes were selected (Figure 8): *SLC2A1, SLC2A3, HK1, HK2, GPI, PFKL, PFKFB3, PFKFB4, ALDOA, ALDOC, GAPDH, ENO1, ENO2, PGK1, PDK1,* and *LDHA* (Appendix A). Under normoxia, seven of the 231 genes, namely, *HK2, PFKFB4*, *ALDOC, PGK1, ENO2, PDK1* and *LDHA* were upregulated by HIF1 and selected for further analysis (Appendix A; Figure 8). Validation of the data from the deep sequencing of glycolysis-associated genes was performed by duplex TaqMan qPCR in the same mRNA samples used for the deep sequencing analysis and additionally for all of the samples cultivated under hypoxic or normoxic conditions as further controls (Figure 1, lane 1–8 and 1*–8*). RT-PCR was used to confirm the differential expression of the selected 16 glycolysis-associated HIF1 target genes identified by deep sequencing under hypoxic conditions (Appendix A) Six of the seven genes were transcriptionally upregulated by HIF1 under normoxic conditions (*HK2, PFKFB4, ALDOC*, ENO2, *PDK1,* and *LDHA*) with the exception of PGK1, were confirmed by qRT-PCR (Figure 9, Appendix A). The results from the qRT-PCR had a 0.9 correlation with the data from the deep sequencing analysis (Appendix A).

### 2.5. Effect of HIF1α and Different Glutamine Levels on Selected Metabolic Products in Two Tumor Cell Lines

In initial experiments with MDA-MB-231 cells using glutamine and/or glucose, it was found that the presence of both (i) a carbon source (glucose) and (ii) a nitrogen source (glutamine) was essential for the rapid proliferation of tumor cells (Figure 10). Therefore, the cells were treated with 11.1 mM glucose and with various concentrations (0–5 mM) of the amino acid glutamine.

The cell lines MDA-MB-231 and XF354 were studied after siRNA-mediated knockdown of HIF1α under normoxia. The proliferation of MDA-MB-231 cells was optimal ≥0.5 mM glutamine, and for XF354 cells the optimal level was the initial concentration of ≥0.1 mM glutamine. Knocking down HIF1α had no significant effect on the proliferation of cells of either lineage (Figure 11a).

The glucose uptake or the release of lactate into the medium from the cells both lineages was not significantly affected by the treatment of different concentrations of glutamine or by HIF1α-specific siRNA (Appendix A).

Furthermore, two waste products of glutamine catabolism, ammonium and alanine, were determined in the medium of the treated cell lines. The release of ammonium was increased at higher glutamine concentrations, but it was not affected by the siRNA-mediated knock down of HIF1α (Appendix A).

The release of alanine was also increased from cells of both lineages with higher glutamine levels. Importantly, at a glutamine level of >1 mM, a significant increase was found in the alanine level released after HIF1α was knocked down (Figure 11b).

### 2.6. Effect of Ascorbic Acid on HIF1α and Select Metabolic Products under Normoxia at Different HIF1α Statuses and with the Addition of Two Different Concentrations of Glutamine

Next, the “therapeutic” effect of ascorbic acid was analyzed in MDA-MB-231 cells that were cultivated under normoxic conditions with 0.25 mM or 5 mM glutamine added in the absence or presence of HIF1α-specific siRNA.

After 72 h of proliferation, the number of cells that were treated with 5 mM glutamine was higher as compared to the number of cells that were treated with 0.25 mM glutamine (Figure 10a and Figure 11a). At low glutamine levels (0.25 mM glutamine), the application of ascorbic acid or siRNA against HIF1α resulted in a significant but minimal reduction in cell number. Such an effect was not found at a higher level of glutamine application (5 mM) (Figure 12a).

The increase in glucose uptake and the release of lactate in the medium was only very limited by the application of ascorbic acid at low concentrations of glutamine (0.25 mM), whereas the knocking down of HIF1α did not have any effect (Figure 12b,c).

Similar to other experiments, the content of the two waste products (ammonium and alanine) of the catabolized amino acid glutamine in the cell culture medium with ascorbic acid added was determined. The release of ammonium was significantly increased due to the effect of ascorbic acid, but only at low levels of glutamine (0.25 mM), and not by the HIF1 siRNA treatment (Figure 12d). Diametrical relationships were found when the release of alanine into the medium was considered. At lower levels of glutamine (0.25 mM), ascorbic acid significantly reduced the amount of alanine released independent of HIF1α silencing, while at high concentrations of glutamine (5 mM), ascorbic acid induced the opposite effect, which increased the amount of alanine release into the medium. This effect was HIF1α-dependent since knocking down HIF1 further increased the concentration of the alanine level released into the medium (Figure 12e).

The ratio of ammonium released to alanine released also depends on the presence of ascorbic acid. At low levels of glutamine, the application of ascorbic acid caused a HIF1α-independent increase in the ratio, which resulted in an increased release of ammonium but low amounts of alanine were detectable. At a high glutamine concentration (5 mM), this correlation appears to represent the inverse ratio (Figure 12f).

## 3. Discussion

### 3.1. Molecular and Biochemical Reasons for the Normoxic HIF1 Activation in Tumor Cells

The catabolism of the carbon source glucose to lactate is a tumor-specific characteristic. Lactate is a significant waste product of the tumor metabolism and is involved in aerobic glycolysis, the so-called Warburg effect [2]. However, glutaminolysis is also a second essential metabolic pathway, because the proliferation of tumor cells requires both a carbon source, such as glucose, and a nitrogen source, such as the amino acid glutamine (Figure 10, Figure 11a and Figure 12a) [13].

In this study, we investigated the causes and consequences of glutamine-induced normoxic HIF1 activity for tumor metabolism. First, we studied the purpose of normoxic HIF activity, which is mainly the process of glutaminolysis. Having previously shown that HIF acts as a pH sensor and pH regulator [13], here we investigated the effect of glutamine/glutaminolysis and various cellular compounds on the stability and activity of HIF1 under normoxia. Second, we investigated the consequences of normoxic HIF activity on tumor metabolism with a focus on the expression of glycolytic and glutaminolysis genes/proteins.

As demonstrated in Figure 1, the α-subunit of the transcription factor HIF1 can be stabilized, even under normoxic conditions [13]. The protein level of CAIX can identify a transcriptional activity of HIF1, which is one major HIF1 regulated protein. However, the underlying molecular mechanisms for the normoxic stabilization and activation of HIF1 are not fully understood.

The generally accepted explanation of HIF1α stabilization is the lack of hydroxylated proline residues in the HIF1α protein, which is usually performed by oxygen-dependent hypoxia-inducible prolyl hydroxylases (EGLNs) [16].

The role of EGLNs can be tested, because EGLNs require ascorbic acid (vitamin C) for their regeneration as well as 2-oxoglutarate and iron for their function [16]. Therefore, we treated the cells with ascorbic acid. As expected, the glutamine-induced normoxic HIF1α protein level completely disappeared in the presence of ascorbic acid (Figure 6). However, this statement is not an entirely valid measure, since normoxic HIF1α stabilization was not induced by the lack of ascorbic acid, but by the application of glutamine (Figure 3). In contrast, under hypoxia, the effect of ascorbic acid was not convincing (Figure 6).

Although Osipyants and colleagues identified ascorbic acid as the substrate of EGLNs [35], the role of EGLNs in normoxic HIF1α detection is highly questionable. Therefore, we suggest another explanation for the effect of ascorbic acid, which ignores the regeneration of prolyl hydroxylases. Instead, we reply on the fact that ascorbic acid can be actively transported into the mitochondrion to prevent mitochondrial membrane depolarization and protect the mitochondrial genome from oxidative damage [36,37,38].

This information allows for us to re-evaluate the observed effect of ascorbic acid on HIF1α. Therefore, we postulate that ascorbic acid can stabilizes the function of the mitochondrion, which is apparently disturbed by glutaminolysis and the release of ammonia (Figure 13). Therefore, the functionality of the mitochondria seems to be important in the generation of the substances that are required for HIF1α degradation. In that context, it is no coincidence that citrate accumulates because the TCA cycle is highly active [1], and HIF1 is usually not detectable during this period of high TCA activity.

A second explanation states that neither 2-oxoglutarate nor oxygen nor the activities of EGLNs are limiting factors for the stabilization of normoxic HIF1α [39], while the availability of Ac-CoA is a limiting factor [39]. Jeong et al. first described an *ARD-1* (NAA10 N(alpha)-acetyltransferase 10, NatA catalytic subunit (NM_003491)) mediated acetylation of Lys532 in HIF1α, which causes the degradation of HIF1α under normoxia [33]. This acetylation necessarily requires the availability of cytosolic Ac-CoA [33]. The acetylation of HIF1 by *ARD* thus induces the degradation of HIF-1 under normoxia, but to a lower level under hypoxia [33]. Recently, Zhang et al. conclusively confirmed the results under hypoxia that were obtained by Jeong and colleagues. They described that the attenuation of HIF-1α acetylation increased the stabilization and transcriptional activity of HIF1α, and even glycolysis in a mouse xenograft model was promoted [40]. In this context, Choudhary et al. described non-enzymatic acetylation of proteins in direct contact with Ac-CoA [41,42,43]. Moreover, the reactivity of Ac-CoA to lysine side chains increased with increasing pH values [41,44,45]. The competitive cross talk between acetylation and ubiquitination can regulate the stability and the subcellular localization of those proteins [41]. Cytosolic Ac-CoA is generated from citrate, which originates from the mitochondria (generated by the active TCA cycle) and it is transported via the citrate-malate shuttle into the cytosol of the cells.

The acetylation of the ODD domain of HIF1α requires a certain level of cytosolic Ac-CoA [33]. We tested this hypothesis and treated cells with sodium citrate or Ac-CoA and found reduced levels of glutamine-induced normoxic HIF1α stabilization (Figure 5; Appendix A).

Moreover, the knock down of two enzymes that are responsible for the generation of cytosolic Ac-CoA (cytosolic enzymes ATP citrate lyase (ACLY)) and the cytosolic form of the acetyl-CoA synthetase ACSS2 [41,46,47]) enables the reduction of the acetylation of HIF1α and caused an accumulation of normoxic HIF1α independent of glutamine/ammonia-induced treatment in different tumor cell lines (Figure 4). ACLY is an enzyme that interconnects glucose and glutamine metabolism [47] and it is essential for tumor metabolism [47,48] as well as tumor proliferation [49]. ACSS2 also appears to be necessary for the metabolism of malignant cells [50].

These experiments support the hypothesis of Jeong et al., which suggested that the acetylation of HIF1α might be necessary for its stabilization [33]. Treatment with acetylated drugs, such as acetylsalicylic acid or ibuprofen, is expected to lead to a decrease in HIF1α levels under normoxia and hypoxia since suppressed acetylation due to reduced cytoplasmic Ac-CoA levels is responsible for the accumulation of HIF1α under hypoxia [40].

Indeed, glutamine-induced normoxic HIF1α accumulation was significantly reduced in the investigated tumor cells that were treated with acetylsalicylic acid or ibuprofen (Figure 3). These results can be explained by the fact that acetylsalicylic acid has the capacity to acetylate the amino group of the lysine side chains in cellular and extracellular proteins [51], and it seems that HIF1α Lys532 is also affected (Figure 3).

On the other hand, the influence of acetylsalicylic acid, ibuprofen, citrate, or Ac-CoA on the hypoxic HIF1α protein in the investigated tumor cell lines only led to a reduced level or no effect (Appendix A). Large amounts of substances are internalized by the cells through phagocytosis and pinocytosis. However, under hypoxia this intake is substantially reduced, which also creates a major problem in the treatment of hypoxic tumors [52,53,54]. However, increasing the level of acetylsalicylic acid or ibuprofen or using very highly acetylated drugs even significantly reduced the level of HIF1α under hypoxia (Appendix A).

These different data convincingly support the postulation of Jeong et al., and Zhang et al., which suggests that the acetylation status of lysine residues is responsible for HIF1α stability [33,40]. These data also indirectly confirm the possible influence of the level of cytoplasmic acetyl-CoA, which is necessary for the acetylation of proteins. Therefore, we continue to assume that a lack of citrate and cytosolic Ac-CoA are both responsible for the stabilization of normoxic and perhaps even hypoxic HIF1α, and thus the transcriptional activation of HIF1.

Based on these facts, we assume that oxygen deprivation cannot be responsible for the occurrence of normoxic HIF1α. This fact necessitates a first paradigm shift; HIF1α/HIF1 is not a specific hypoxic molecular factor [31,32].

Moreover, we can now propose a second paradigm shift: HIF1α/HIF1 is also stabilized due to the reduced acetylation of lysine residues, and the acetylation of Lys532 of HIF1α/HIF1 leads to its degradation.

### 3.2. Consequences of Glutamine-Induced Normoxic HIF1 Activity for the Tumor Glycolysis/Metabolism

Therefore, we first studied the commonalities and differences in the transcriptional activation of genes by HIF1 under hypoxia and normoxia while using deep sequencing data for glycolysis-associated genes. As expected, we found that hypoxic HIF1 significantly upregulated 16 genes that are involved in glycolysis or that block the TCA cycle (PDK and LDHA) at the mRNA level (Appendix A, Figure 8; Appendix A).

The activity of normoxic HIF1 is not well understood [16,21,22,23,24,25,26,55]. However, our data show that, even under normoxic conditions, six of the 16 hypoxic glycolysis-associated genes (HK2, PFKFB4, ALDOC, ENO2, PDK1, and LDHA) were significantly upregulated (and log2-fold change ≥1) due to the glutamine-induced activity of normoxic HIF1 (Appendix A; Figure 9).

The knocking down of HIF1α had no significant effect on the uptake of glucose or the release of lactate (Appendix A). Other authors found an effect of hypoxic HIF1α on glycolysis in a mouse xenograft [40] and an effect of HIF1α on glucose flow and lactate release under aerobic and hypoxic conditions [59].

However, the knockdown of HIF1α affected the ammonia/alanine level, whereby a significantly increased level of alanine was released (Figure 11b). This is interesting, since the amino acid alanine is similar to ammonia, a waste product of glutaminolysis, and both waste products are released by tumor cells at a 1:1 ratio [8]. As part of the Cahill cycle [60,61], which is similar to the Cori cycle (for lactate), ammonia binds to pyruvate to form alanine. Subsequently, alanine is released into the cell environment/blood, where it is used in gluconeogenesis by the liver [60,61]. Additional ascorbic acid reduces the level of released alanine at low glutamine concentrations (when normoxic HIF1α is not present [13]) in our experiments. Thus, the effect of ascorbic acid is independent of HIF1 status. It seems that HIF1 maintains a low level of released alanine, but only in the presence of high levels of glutamine/ammonia (Figure 11b, Figure 12e), because only under these conditions is normoxic HIF1 present [13]. The effect of ascorbic acid is reversed at high levels of glutamine/ammonium (Figure 12d), when normoxic HIF1α is usually induced (Figure 2). At 5 mM glutamine and after downregulation of HIF1α using ascorbic acid, or siRNA or the combination of both, a significantly increased level of alanine is released (Figure 12e). Thus, the function of HIF1 does not seem to include the amination of pyruvate, which generates alanine. The alternative regeneration of reduction equivalents through a reduction of pyruvate into lactate is more important in a “stressful” environment, when HIF1 is usually active. Moreover, we postulate that the release of increased levels of alanine, as mediated by knocking down HIF1α, is likely due to the higher concentrations of pyruvate and ammonia inside the cell.

### 3.3. Consequences for A New Interpretation of the Warburg Effect

A possible role of HIF1 in glycolysis and the Warburg effect (aerobic glycolysis) has been already postulated several times by various groups investigating hypoxic HIF1 [28,59,62]. This suggestion is understandable, since HIF1 regulates the genes of glycolysis, PDK1, and LDHA, which are also essential genes for the Warburg effect. Denko noticed that “HIF1 alone can drive the major metabolic changes within the tumour that were identified by Otto Warburg” [63].

However, at this point it must be asked how it is possible that some authors recognize a role of HIF1 in the Warburg effect, when the Warburg effect is undoubtedly a normoxic phenomenon? Thus, as a first step, the existence and activity of normoxic HIF1/HIF1α should be detected.

We found that six of 16 HIF1-linked glycolysis-associated genes were also upregulated by normoxic HIF1 (e.g., PDK1 and LDHA) (Figure 8 and Figure 9).

However, it is highly unlikely that the deregulation of a single transcription factor, such as HIF1, could cause the Warburg effect. To this effect, Denko postulated that “the reduced mitochondrial function could be responsible for the tumor cell growth advantage” [63].

The proliferation of tumor cells requires both a carbon source (glucose) and a nitrogen source (e.g., glutamine) (Figure 10 and Figure 11a) [6,7,8,9,30]. We found that tumor cells had the maximum in vitro proliferation rates (Figure 11a) at physiological glutamine levels (in human blood 0.5–0.9 mM) [64].

This fact suggests that a third paradigm shift is necessary: The Warburg effect, as an effect of rapidly proliferating cells [4], is impossible without a nitrogen source such as glutamine.

However, the catabolism of amino acids, such as glutamine, results in the release of ammonia/ammonium, which is capable of activating PDK1 and shutting off the TCA cycle [57]. Without functional mitochondria with an operational TCA cycle, glycolysis (the Warburg effect) is the only alternative way to break down glucose and regenerate the reduction equivalents.

Thus, the localization of glutaminolysis within the mitochondria seems to be simultaneously necessary and problematic for tumor cells. Tumor cells use the mitochondria to a greater extent in the production of important metabolites out of glutamine (for generation of other amino acids or membranes) for proliferation [30]. Therefore, the TCA cycle could be inoperable in a toxic environment due to the ammonia/ammonium released during glutaminolysis.

We must conclude that the activation of HIF1, whether being hypoxic or normoxic, is merely the result of malfunctioning mitochondria (in term of the TCA cycle). Under hypoxia, the lack of oxygen and under normoxia, the effect of ammonia/ammonium can bring the TCA cycle to a standstill [57]. However, this level of TCA cycle dysfunction would affect overall mitochondrial metabolism (e.g., generation of citrate), and even the functional integrity of the mitochondria.

The obscure association of glycolysis (Warburg effect) and the release of ammonia was previously described by Otto Warburg and colleagues, when they found that 1 M ammonia was released for every 8 M of oxygen consumed when the tissue was treated with a glucose-free solution [65]. If the release of ammonia correlates with glycolysis, then the reason for the Warburg effect (which is, indeed enhanced glycolysis) must be to poison the essential pathways in the mitochondrion, which is already explained by the effect of ammonia/ammonium on PDK1 and the TCA cycle [57]. Reduced mitochondrial function and, as a result, increased glucose uptake, was described for aerobic glycolysis [63]. The activity of normoxic HIF1, triggered by ammonia/ammonium, contributes to the stabilization of this non-physiological condition.

Thus, HIF1 activity can contribute to the stabilization of metabolism that is out of balance and invoke a new equilibrium. Moreover, Meng et al., concluded, “The Warburg effect may be the metabolic consequences secondary to the nitrogen anabolism” [12], which, at least, supports our interpretation.

This model can be further supported by the fact that HIF1 can be stabilized and activated due to increased levels of metabolites, such as succinate or fumarate, when some TCA enzymes are mutated. Thus, the flow of metabolites is blocked, and succinate and/or fumarate accumulates, which also leads to the discontinuation of the TCA cycle [63].

Finally, these facts necessitate a fourth paradigm shift: The Warburg effect could be caused by the poisoning of mitochondria via the metabolite ammonia/ammonium.

### 3.4. Therapeutic Consequences

Chemical compounds with acetylation ability, such as ibuprofen and acetylsalicylic acid, influence HIF1α acetylation levels (Figure 3), which can contribute to the degradation of this protein [33,40]. The effect of acetylsalicylic acid is also of interest, because a tumor-associated benefit of this drug has already been described.

It has been shown in eight studies of different tumorous entities that aspirin (acetylsalicylic acid) can delay tumor-associated death (reviewed in Rothwell et al., 2011) [66]. However, the latent period before the effect on deaths was found to be approximately five years for esophageal, pancreatic, brain, and lung cancers, and was even more delayed for gastric, colon, and prostate cancers [67,68]. This unexpected, but the useful side effects of aspirin can be explained by its potential to acetylate other amino acid side chains, e.g., lysines of other proteins [51]. This side effect of aspirin seems to also decrease the stability of HIF1 (Figure 3), which is essential for tumor cells stabilizing the misbalances of tumor metabolism [13].

The more surprising finding is the impact of ascorbic acid in this context. Similar to the treatment of acetylsalicylic acid, we found an effect of ascorbic acid on the normoxic destabilization of glutamine-induced HIF1α (Figure 6). We interpreted this effect, not as a possible result of regenerated EGLNs, but rather as an outcome of the direct ascorbic acid influence on mitochondrial activity and functionality.

However, in the nitrogen metabolism of cells, there is an apparent effect of ascorbic acid (regardless of its impact on HIF1α) (Figure 12d,e) [69,70] at physiological glutamine levels. Simultaneously, a significantly greater amount of ammonium is released as a significantly reduced amount of alanine is released from the same tumor cells (Figure 12d,e).

In addition, these data show a possible significant effect of ascorbic acid on the nitrogen metabolism (Figure 12d,e). The results of Kuiper et al., [69] from an in vivo study, suggest that ascorbic acid reduced the HIF1α protein level in the colorectal tumors and this reduction was correlated with longer survival of patients. This finding indicates that a state of pseudohypoxia after prior induction of HIF1 levels by, e.g., metabolism of amino acids, can be treated in vivo, e.g., by use of ascorbic acid (Figure 6) [69]. This interpretation is reasonable, because hypoxic HIF1α cannot be destabilized by ascorbic acid (Figure 6; Appendix A). Moreover, the effect of ascorbic acid on the Warburg effect has even been discussed [71].

For this reason, more extensive investigations are necessary. In addition, supporting tumor therapy with ascorbic acid and acetylsalicylic acid could be useful for normoxic and hypoxic tumors (Appendix A).

Another approach in terms of a tumor therapeutic strategy is to affect the glutamine uptake or glutamine metabolism/glutaminolysis. It is surprising that tumor cells use the amino acid glutamine as a metabolite, which otherwise serves as a nitrogen transporter [8,30]. Extensive glutaminolysis does not occur in normal tissue, except in rapidly proliferating normal tissue (wound healing and immune responses) [4,72,73,74]. A therapeutic opportunity could be created from the excessive consumption of glutamine by tumor cells [30,75,76]. Targeting glutamine consumption could be a metabolic approach for the treatment of cancer, as described in review articles [30,77].

Glutamine uptake can be reduced by glutamine receptor inhibitors, and glutaminolysis can be inhibited by the inhibition of glutaminases (GLSs), which catalyze the conversion of glutamine to glutamate in mitochondria (reviewed in Jin et al., 2016) [77]. Recently, it was reported that the GLS inhibitor CB-839 might increase the sensitivity of ovarian carcinoma cells to PI3K/mTOR inhibition [78], and several Phase I CB-839 clinical trials are listed (www.clinicaltrials.gov, last accessed on: 17 September 2019).

Regarding the key factor that is involved in tumor progression, metastasis, appears to be directly related to HIF1 and aerobic glycolysis [1,79,80]. Based on this finding, the precise biological characterization of both HIF1 and aerobic glycolysis and glutaminolysis, including ammonium/ammonia release, are important in the search for new therapeutics against cancer and metastasis.

There are also clinical studies that aim to inhibit HIF1α in cancer therapy [81]. Bortezomib, a proteasome inhibitor that was approved by the FDA for multiple myeloma treatment in 2008, indirectly inhibits the transcriptional activity of HIF1 and even causes radiosensitivity in tumor cells [82]. More directly, EZN-2968, an anti-HIF1α LNA AS-ODN, was recently tested in clinical phase I studies for different cancers (www.clinicaltrials.gov, last accessed on: 17 September 2019). Although, targeting HIF1α directly or indirectly appears to be promising for future cancer therapy; the results of further clinical studies targeting HIF1α must be first made available.

It is conceivable that the glutaminolysis of tumors could be influenced and that the administration of additional glutamine could support the immune system of patients. By disrupting the regulatory detoxification system of tumor cells via the destabilization of HIFs and the pH of the micro milieu of the tumor, e.g., by specific drugs (acetylation drugs or ascorbic acid) the specific homeostasis of the tumors could be disturbed.

Finally, a diet of reduced carbon might possibly support tumor therapy and it should be investigated in the context of the findings of this study. The high consumption of glucose by tumor cells could be driven by the need to compensate for the negative effect of catabolized amino acids, such as glutamine, making dietary approaches useful in supportive therapeutics (but always under medical observation).

## 4. Materials and Methods

### 4.1. Cell Culture and Treatment

The human breast cancer cell line MDA-MB-231 (ATCC, Rockville, MD, USA) was cultured in monolayers in RPMI 1640 medium (Lonza, Walkersville, MD, USA) containing 10% fetal bovine serum (FBS) (Capricorn, Ebsdorfergrund, Germany), 185 U/mL penicillin (Invitrogen, Karlsruhe, Germany), and 185 μg/mL streptomycin (Invitrogen, Karlsruhe, Germany) at 37°C in a humidified atmosphere supplemented with 5% CO_2._ For additional experiments, we analyzed the human breast cancer cell line MCF-7, the human glioma cell line U251MG, the human osteosarcoma cell line Saos-2, and the human colorectal carcinoma cell line HCT 116, which were obtained from the American Type Culture Collection (ATCC, Rockville, MD, USA). The XF354 cell line was derived from a primary SCC from the floor of the mouth (Deutsches Krebsforschungszentrum, Heidelberg, Germany) and the CAL-33 cell line (derived from a primary tumor of the tongue; Deutsche Sammlung von Mikroorganismen und Zellkulturen GmbH, Braunschweig, Germany) were cultivated under identical conditions.

The cells were treated with different levels of L-glutamine (Sigma, Steinheim, Germany). All of the experiments on the effect of glutamine were performed while using phenol red-free RPMI 1640 (PAA, Pasching, Austria, and Life Technologies, Darmstadt, Germany) without glutamine supplementation. The experiments were typically performed under normoxic conditions (21% oxygen). Hypoxia (<1% oxygen) was achieved while using a gas generator system, as previously described [83].

Moreover, for experiments, the cells were treated with 3 mM ibuprofen Caesar and Loretz (Caesar and Loretz, Hilden, Germany) dissolved in DMSO or with 10 mM acetylsalicyl acid (dissolved in DMSO (Sigma, Steinheim, Germany) or as Aspirin I.V. 500 mg; Bayer, Leverkusen Germany) under normoxic or hypoxic conditions in RPMI medium containing glutamine, 10% FBS, and 1 mM sodium pyruvate (Invitrogen, Karlsruhe, Germany). The cells were treated additionally with phenylacetate ester, phenylacetate, Ac-CoA (Sigma, Steinheim, Germany) or sodium citrate, ammonium chloride (Roth, Karlsruhe, Germany). The cells were harvested 24 h after the start of the experiment for detection of HIF1α- protein levels.

### 4.2. Transfection

The cells were transfected with ACLY-, ACSS2-, HIF1α-specific siRNA (Life Technologies, Darmstadt, Germany) or with transfection reagent without siRNA while using the INTERFERin reagent (Polyplus Transfection, Illkirch, France) according to the manufacturers’ instructions (please see also Appendix A) see also [13]. For the experiment shown in Figure 1 and the deep sequencing data, the cells were then treated in (a) RPMI medium without glutamine and without FBS, (b) RPMI medium without glutamine, but with 1% FBS, (c) RPMI medium with 5 mM glutamine, but without FBS, and (d) RPMI medium with 5 mM glutamine and 1% FBS. The experiments were performed under normoxic (21% oxygen) or hypoxic (0.1% oxygen) conditions. The experiments were performed at least three times. The samples were harvested 24 h after the start of the experiments (Figure 1).

### 4.3. Cell Proliferation Experiments

MDA-MB-231 cell proliferation was assayed in at least five independent experiments (Figure 10). One million cells were grown in RPMI 1640 medium (Lonza, Walkersville, MD, USA) with 2.2 mM glutamine, 11.1 mM glycose containing 10% FBS, 185 U/mL penicillin, and 185 μg/mL streptomycin at 37 °C in a humidified atmosphere supplemented with 5% CO_2_ overnight. Subsequently, the cells were cultivated in DMEM (without glucose and without glutamine) containing 10% FBS, 185 U/mL penicillin and 185 μg/mL streptomycin. As shown in Figure 10, cells were either cultivated (1) without glucose and without glutamine, (2) 5 mM glutamine, (3) 11.1 mM glucose, 4) 5 mM glutamine and 11.1 mM glucose (Merck, Darmstadt, Germany) for 72 h. Afterwards, the cells were rinsed in PBS, trypsinized, and counted with a Z1 cell counter (Beckman Coulter, Krefeld, Germany).

For further experiments (Figure 11 and Figure 12), MDA-MB-231 cells and XF354 cells were transfected with 10 nM Silencer Select siRNA (ID: s6539), HIF1α-specific siRNA (Life Technologies, Darmstadt, Germany) overnight while using the INTERFERin reagent (Polyplus Transfection, Illkirch, France). The control cells were transfected with INTERFERin reagent alone while using RPMI 1640 medium with glutamine, 10% FBS, and 1 mM pyruvate. The next day, 200,000 cells were seeded in a new flash using RPMI 1640 medium with glutamine, 10% FKS and 2.2 mM glutamine and 1 mM pyruvate (Invitrogen, Karlsruhe, Germany), 185 U/mL penicillin(Invitrogen, Karlsruhe, Germany), and 185 μg/mL streptomycin (Invitrogen, Karlsruhe, Germany). After 8 h, the medium was replaced by RPMI 1640 RPMI without glutamine (Life Technologies, Darmstadt, Germany as well as 0–5 mM L-glutamine (Sigma, Steinheim, Germany), and (Figure 11 and Figure 12) 100 µM ascorbic acid (Amos Vital GmbH, Köln, Germany) or 0–10 mM NH_4_CL (Roth, Karlsruhe, Germany) (Appendix A). The XF 354 cells were cultured and treated in a manner that is similar to that of the MDA cells in DMEM, no glucose, no glutamine, and no phenol red (Life Technologies, Darmstadt, Germany) (additional 11 mM glucose (Merck, Darmstadt, Germany) and only 2% serum (Capricorn, Ebsdorfergrund, Germany) was used). After 72 h, the cells were counted and were isolated for RNA and protein generation. The supernatant (3.5 mL) was stored at 4 °C or for a longer time at –20 °C, as was the control medium, which was not used for cell cultivation.

### 4.4. Metabolic Assay

The medium that was used in the MDA-MB-231 and XF354 cell experiments of Figure 11 and Figure 12 was also used to calculate the glucose, lactate, ammonium, and alanine concentrations.

The glucose concentration of the supernatant was measured while using a glucose (HK) assay kit (Sigma, Steinheim, Germany)—modified according to the manufacturer’s instructions using a glucose standard solution (Sigma, Steinheim, Germany).

The lactate concentration of the supernatant was measured using a lactate assay kit (Trinity Biotech, Bray, Ireland)—modified according to the manufacturer’s instructions using a lactate standard solution (Trinity Biotech, Bray, Ireland).

The ammonium concentration of the supernatant was measured using an ammonia assay kit (Sigma, Steinheim, Germany)—modified according to the manufacturer’s instructions while using an ammonia standard solution (Sigma, Steinheim, Germany).

The alanine concentration of the supernatant was determined using the Amplite ^TM^ colorimetric L-alanin assay kit (AAT BioQuest Inc. Sunnyvale USA)—modified according to the manufacturer’s instructions using an alanine standard solution (Sigma, Steinheim, Germany).

The concentrations of glucose, lactate, ammonium and alanine level were calculated using the standard curve, and residual medium served as a reference and to estimate the glutamine decay in the medium (for the ammonium measurement only).

### 4.5. Western Blot Analysis

Western blot analysis was performed as previously described [83]. Briefly, the cells were treated with lysis buffer (50 mM Tris-HCl, 200 mM NaCl, 1 mM EDTA, 1 mM EGTA, 1% Triton X-100, 0.25% deoxycholate, and protease and phosphatase inhibitors) and homogenized by sonication. Subsequently, the protein concentration was determined while using the Bradford method. The protein solution was separated using 4–12% Bis-Tris gradient gel (Life Technologies, Darmstadt, Germany) by electrophoresis and then transferred onto an Immobilon PVDF membrane (Millipore GmbH, Schwalbach, Germany). Subsequently, the membranes were incubated at 4°C overnight in TBS (50 mM NaCl, 30 mMTris-HCl pH 7.5 with 10% nonfat milk (Carl Roth, Karlsruhe, Germany) with the primary anti-β-actin monoclonal antibody (mAb) (Clone AC-15, dilution 1:3000, host mouse) from Sigma (Steinheim, Germany), the anti-CAIX mAb (Clone M75, dilution 1:2000, host mouse) from Bioscience (Bratislava, Slovak Republic), the anti-ACSS2-antibody (ab66038, dilution 1:1000, host rabbit) from abcam (Abcam, Cambridge, UK), the anti ACLY-antibody (Phospho-ATP-Citrate Lyase (Ser455)) (#4331, dilution 1:1000, host rabbit) (Cell Signaling Technology, Danvers, MA, USA), or anti-HIF1α antibody (dilution 1:2000, host mouse) from BD Bioscience (BD Bioscience, Heidelberg, Germany). The PVDF-membranes (Millipore GmbH, Schwalbach, Germany) were washed three times with TBS and incubated for 1 h with HRP-conjugated secondary antibodies (1:5000, DAKO, Hamburg, Germany). The membranes were then treated with ECL or ECL Prime Western Blotting Detection reagents (GE Healthcare, Little Chalfont, UK) and then evaluated by a FUSION-FX7 SPECTRA chemiluminescence-fluorescence-imaging System (Vilber Lourmat, Eberhardzell, Germany), CL-XPosure™Film (Agfa HealthCare, Mortsel, Belgium) or ChemiDoc™ Touch Imaging System (BioRad, Hercules, CA, USA) for detecting chemifluorescence signals.

### 4.6. RNA Preparation, cDNA Synthesis and Quantitative Real-Time PCR (qRT-PCR)

The RNA was isolated while using a ZR RNA MiniPrep kit (Zymo Research, Irvine, USA) according to the manufacturer’s instructions. RNA concentration was assessed on a Nanodrop 2000C (NanoDrop products, Wilmington, DE, USA). The RNA samples were stored at −80 °C. For cDNA synthesis, a RevertAid™ H Minus First Strand cDNA Synthesis kit (Fermentas, St. Leon-Rot, Germany) was used according to the manufacturer’s instructions. PCR was performed while using a Rotor-Gene RG-6000 (LTF, Wasserburg, Germany) and duplex (gene of interest (FAM labeled probe reference (RPL9) VIC labeled probe) (Applied Biosystems, Foster City, CA, USA). For the quantification of the transcript amounts, the ΔΔCt method was applied.

### 4.7. Deep Sequencing

For deep sequencing, RNA from MDA-MB-231 cells that were obtained after the different treatments was used in four independent experiments. A total of 500 ng/total RNA was used for library synthesis with the TrueSeq RNA Sample Prep Kit v2 (Illumina, San Diego, CA, USA) according to the manufacturer’s protocol. The barcoded libraries were purified and quantified while using a Library Quantification kit–Illumina/Universal (KAPA Biosystems) on a TaqMan 7500 Real-Time PCR System. A pool of up to 10 libraries was used for cluster generation at a concentration of 10 nM each using an Illumina cBot. The sequencing of 2 × 100 bp was performed with an IlluminaHighScan-SQ sequencer at the sequencing core facility of the Faculty of Medicine (University Leipzig) using version 3 chemistry and flowcell according to the instructions of the manufacturer.

### 4.8. Intracellular pH Measurement

1.5 × 10^4^ cells were incubated with RPMI 1640 (containing L-glutamine, D (+)—glucose, phenol red) supplemented with 1% FBS in 1.5 mL total volume and cultivated overnight at 37 °C., 5% CO_2_, 21% O_2_ on coverslips in six-well plates. On the following day, the coverslips in the six-well plates were overlaid with 5 mL RPMI 1640 (without L-glutamine, without phenol red), supplemented with 5% FCS and 5 mM L-glutamine and incubated for 24 h at 37 °C, 5% CO_2_, and 21% O_2_. The next day, the cells were treated with 5 µM of the fluorescent dye BCECF (2’,7’-Bis-(2-carboxyethyl)-5,6-carboxyfluorescein) (Life Technologies) in 1 mL medium for 15 min., and then washed twice. The BCECF treated cells were placed on the microscope (Axiovert S100, Zeiss, Jena, Germany) and an inflow of the preheated buffer solutions with different pH values or media (with or without 5 mM Nh_4_Cl of a known pH value) is realized via a reservoir. Subsequently, ≥14 vital cells were selected and a region of interest (ROI) is defined as a measuring point in the cytoplasm via PC. The measurement interval (10 s/5 s) and the exposure time (334 ms/223 ms) are set.

The sequence of excitation wavelengths 450nm and 490nm is automatically controlled by a shutter system. The fluorescence is detected via lens (Fluar 40×/1.30 Oil) and an optical filter (535–550 nm) from the camera system. The measurement was performed by an established detection system in the Julius-Bernstein-Institute (Medical Faculty, Martin Luther University Halle-Wittenberg, Germany).

### 4.9. Bioinformatics Approach and Statistical Methods

We set up a Snakemake [84] workflow, called ‘VipeR_HIF1alpha’, for the analysis of the RNA-Seq and qPCR data in a reproducible, automated, and partially contained manner. VipeR_HIF1alpha (https://github.com/GrosseLab/VipeR_HIF1alpha, last accessed on: 16 September 2019) is implemented so that alternative or similar analysis can be added or removed. The RNA-Seq paired-end reads were adaptor clipped using cutadapt [85] and quality filtered using sickle [86]. Filtered reads were mapped with STAR [87] to the human genome (GRCh38.82), and based on that mapping transcript counts were quantified with salmon [88]. These transcript counts were summarized to gene counts with tximport [89]. A gene was defined as expressed if it exhibited more than one counts in at least two out of four biological replicates in at least one treatment. Genes which were not expressed according to our definition were removed from the analysis. Hence, we also kept genes in the analysis showing expression in just one treatment. Integrated normalization and differential expression analysis were conducted with edegR [90]. The genes were declared as differentially expressed according to a false discovery rate (FDR) below 0.05 obtained by the Benjamin-Hochburg procedure, and when the log2-fold change was greater than one or smaller than minus one. Further, the Database for Annotation, Visualization, and Integrated Discovery DAVID [91] were used for functional annotation of the differentially expressed genes. Normalized qPCR expression values for normoxic and hypoxic treatments are fitted gene-wise with a linear model with three factors (siRNA, glutamine (G), and serum (S)) by using R package limma [92] with lm(qPCR ~ siRNA + G + S + siRNA:G + siRNA:S + G:S + siRNA:G:S). The analysis of variance (ANOVA) for the fitted model was computed by using the ‘aov’-function of the R package stats [93]. Afterward, a Post hoc analysis was performed by using Tukey’s ‘Honest Significant Difference’ method to assess significantly different levels. A significance level of 0.05 was used for statistical analysis for four independent experiments (Appendix A). We used an FDR cutoff value of 0.05 (Appendix A).

## 5. Conclusions

This manuscript highlights the effect of the amino acid glutamine and its metabolism on the detection of normoxic HIF1α. In particular, the acetylation of HIF1α appears to be necessary for stabilization under normoxia, which can be influenced by the administration of e.g., acetylsalicylic acid or ascorbic acid. Normoxic HIF can upregulate numerous genes, including some genes involved in glycolysis. However, HIF1 also appears to be able to actively affect nitrogen metabolism, especially that of the amino acid alanine. The role of HIF1 in stabilizing pH (via CAIX/XII) for the processing of the degradation product of glutaminolysis (ammonia/ammonium) was addressed, as was its possible role in the Warburg effect. In our opinion, HIF1 has a metabolically stabilizing role. The cause of the Warburg effect is probably not caused by the deregulation of HIF1. Rather, it is attributable to the inhibition of the functionality and perhaps even the integrity of the mitochondria in tumor cells. A very negative influence of ammonia/ammonium on mitochondria can be assumed and could be one reason for aerobic glycolysis. All of these effects cannot be individually considered, and only the consideration of the overall picture of this regulated metabolic complex provides an approach to understanding tumor-specific metabolism.

## Figures and Tables

**Figure 1 ijms-20-04742-f001:**
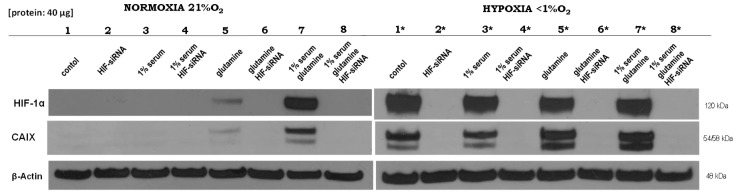
Altered HIF1α-dependent effects of normoxia and hypoxia under different culture conditions. Western blot analysis was performed 24 h after the application of different treatments for HIF1α and carbonic anhydrase (CAIX), while β-actin served as a control. (Lane 1) Medium control; (lane 2) siRNA against HIF1α; (lane 3) 1% fetal bovine serum; (lane 4) 1% fetal bovine serum and siRNA against HIF1α; (lane 5) 5 mM glutamine; (lane 6) 5 mM glutamine and siRNA against HIF1α; (lane 7) 1% fetal bovine serum and 5 mM glutamine; and, (lane 8) 1% fetal bovine serum, 5 mM glutamine and siRNA against HIF1α under normoxic or hypoxic (*) conditions in MDA-MB-231 cells. The experiments were performed in RPMI 1640 in the presence of 11.1 mM glucose, but without glutamine. Deep sequencing analysis was performed for each of the corresponding RNA samples (lanes 7, 8, 7*, and 8*) from four independent experiments (Please see Appendix A and the results from the densitometric evaluation of the Western blots of four independent experiments).

**Figure 2 ijms-20-04742-f002:**
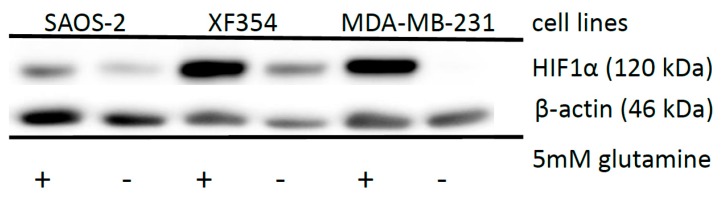
Effects of glutamine application on HIF1α expression in different tumor cell lines. Saos-2, XF354 and MDA-MB-231 cells were left untreated or treated with 5 mM glutamine in RPMI 1640 medium without glutamine under normoxic conditions for 24 h followed by Western blot analysis of HIF1α and β-actin as a loading control. Data from three independent experiments (Please see Appendix A and the results from the densitometric evaluation of the Western blots).

**Figure 3 ijms-20-04742-f003:**
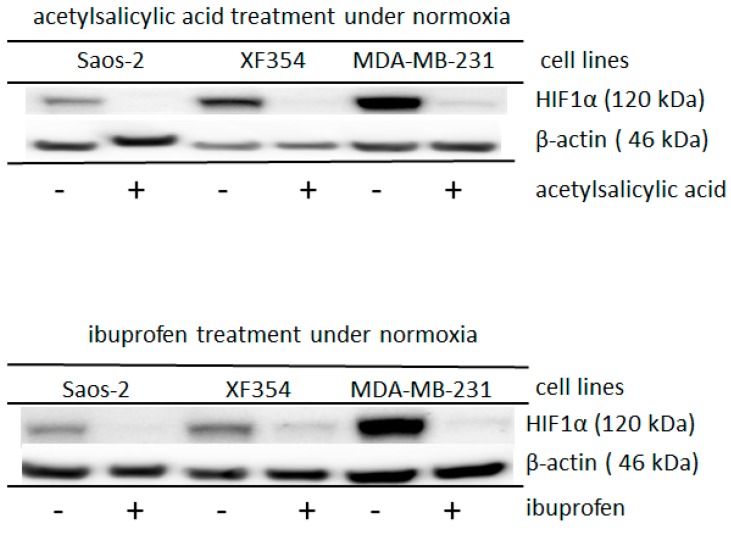
Altered HIF1α expression after the application of acetylsalicylic acid or ibuprofen in different cell lines under normoxia. Saos-2, XF354 and MDA-MB-231 cells were treated with 10 mM acetylsalicylic acid or 3 mM ibuprofen in RPMI 1640 medium with 5 mM glutamine under normoxic conditions for 24 h. Western blot analysis was performed with anti-HIF1α monoclonal antibody and anti-β-actin monoclonal antibody as a control. Acetylsalicylic acid or ibuprofen significantly reduced the glutamine-induced expression level of the normoxic HIF1α. Data from three independent experiments (Please see Appendix A and for the densitometric evaluation of the Western blots). For the results of experiments under hypoxic conditions, please see Appendix A.

**Figure 4 ijms-20-04742-f004:**
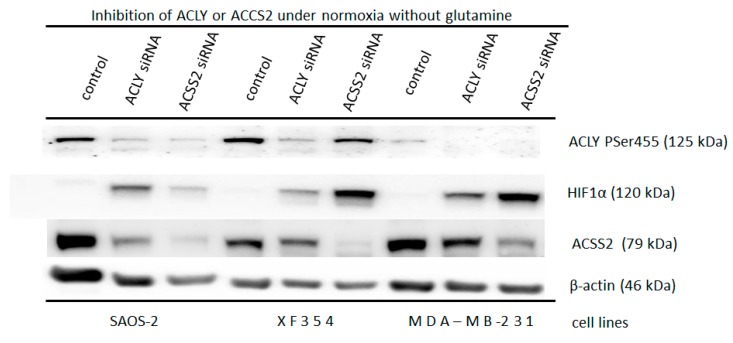
Glutamine independent induction of normoxic HIF1α via down regulation of ACLY or ACSS2. Application of 20 nM siRNA against ATP citrate lyase (ACLY) or the cytosolic form of the acetyl-CoA synthetase (ACSS2) in Saos-2, XF354 and MDA-MB-231 cells in RPMI 1640 medium without glutamine under normoxic conditions for 48 h. This was followed by Western blot analysis of HIF1α, ACLY (Ser 455), and ACSS2 and β-actin as a loading control. Data from three independent experiments (Please see Appendix A and the results from the densitometric evaluation of the Western blots).

**Figure 5 ijms-20-04742-f005:**
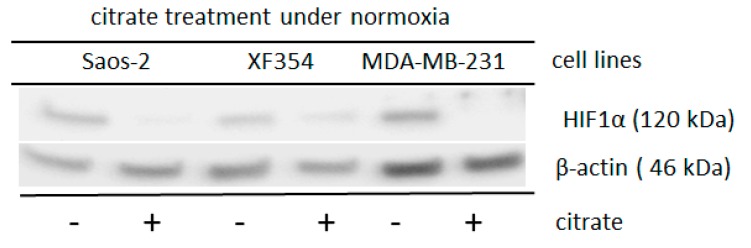
Altered HIF1α expression after the application of sodium citrate in cells of different lineages under normoxia after simultaneously treatment with glutamine. Saos-2, XF354 and MDA-MB-231 cells were treated with 8 mM (the XF354 cells were treated with only 4 mM sodium citrate) in RPMI 1640 medium with 5 mM glutamine under normoxic conditions for 24 h. Western blot analysis was performed with anti-HIF1α monoclonal antibody and anti-β-actin monoclonal antibody as a control. Sodium citrate significantly reduces the glutamine-induced normoxic HIF1 α-level. Data are from three independent experiments (Please see Appendix A and the results from the densitometric evaluation of the Western blots).

**Figure 6 ijms-20-04742-f006:**
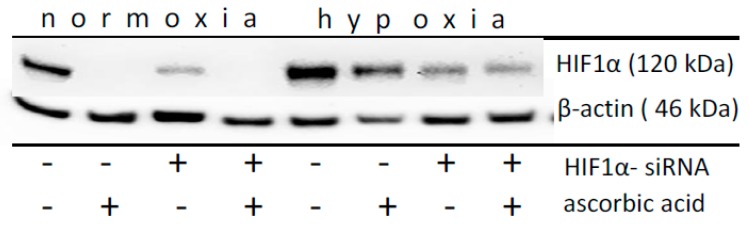
Impact of ascorbic acid on the normoxic and hypoxic HIF1α level in MDA-MB-231 cell. After the of application of HIF1α-siRNA and/or the simultaneously application of 5 mM glutamine and/or 100 µM ascorbic acid for 24 h under normoxic or hypoxic conditions, western blot analysis of HIF1α and β-actin as a loading control was performed. Ascorbic acid significantly reduces the glutamine-induced normoxic HIF1 α-level. Data from three independent experiments. (Please see Appendix A and the results from the densitometric evaluation of the Western blots).

**Figure 7 ijms-20-04742-f007:**
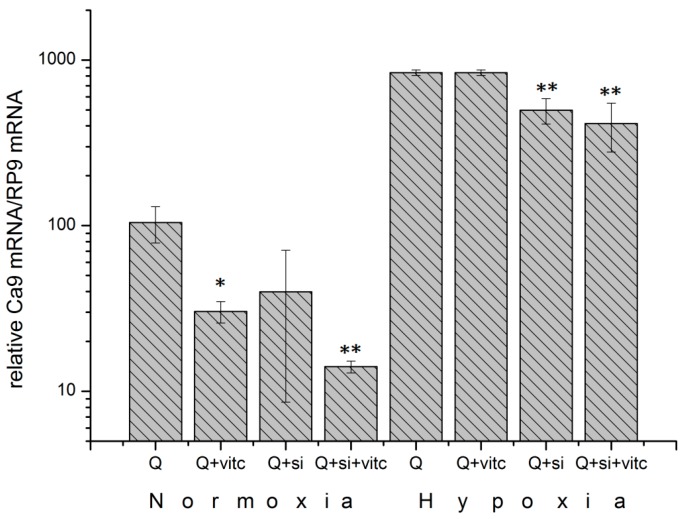
Reduction of relative *CA9* mRNA expression under normoxia and hypoxia (<0.1% oxygen) 24 h after the application of 5 mM glutamine (Q) and 100 µM ascorbic acid (vitc) or HIF1α siRNA (si) in MDA-MB-231 cell. Data correspond to the protein data presented in Figure 6 and demonstrate the transcriptional activity of HIF1 under the influence of ascorbic acid (vitc). Ascorbic acid significantly reduces the *CA9* mRNA expression under normoxia. (* −*p*-value < 0.05; ** −*p*-value < 0.01, Student *t*-test as compared to control (Q)).

**Figure 8 ijms-20-04742-f008:**
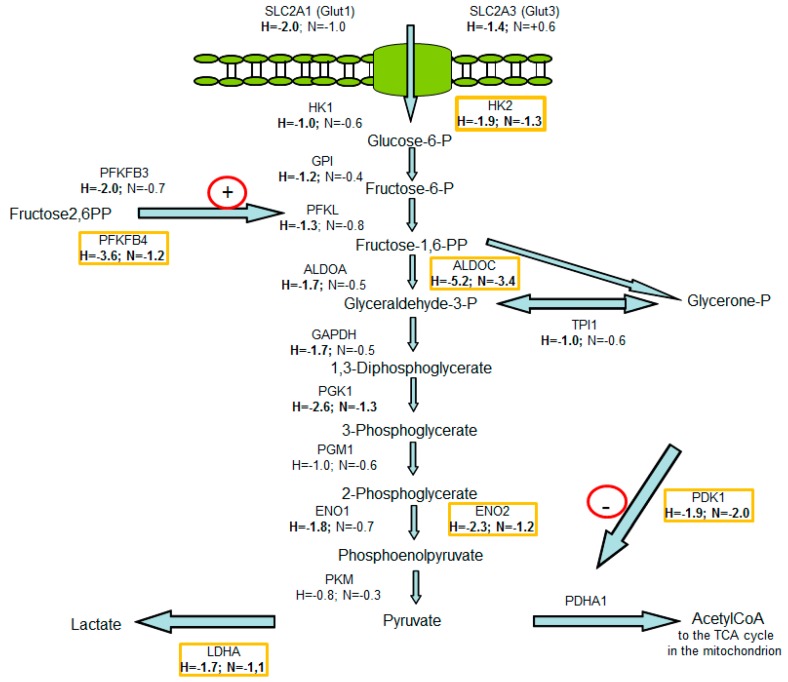
Gene mRNA levels using deep sequencing data of genes involved in glycolysis after HIF1α was knocked down in the MDA-MB-231 cells. RNA was harvested 24 h after the start of the experiments. The experiments were performed in RPMI 1640 medium without glutamine but with 11.1 mM glucose. In this scheme, the mRNA levels are depicted as a ratio of untreated cells to HIF-siRNA transfected cells either under normoxic (N) or under hypoxic (H) conditions. Deep sequencing analysis was performed while using the hypoxic samples after the application of 5 mM L-glutamine and 1% fetal bovine serum (corresponding to lane 7* in Figure 1) and hypoxic samples that were additionally treated with HIF1α-specific siRNA (corresponding to lane 8* in Figure 1) in four independent experiments. H (hypoxia)-describes the quotients of the transcript level of genes from the hypoxic HIF1-positive samples (corresponding to lane 7* in Figure 1) divided by the transcript level from the hypoxic HIF1-negative samples (corresponding to lane 8* in Figure 1). N (normoxia) describes quotients of the transcript level of the normoxic HIF1-positive samples (corresponding to lane 7 in Figure 1) divided by the transcript level of the normoxic HIF1-negative samples (corresponding to lane 8 in Figure 1). Genes marked yellow were significantly upregulated by HIF1 under normoxic conditions, as was validated using qPCR (Appendix A). Figure was adapted from [34].

**Figure 9 ijms-20-04742-f009:**
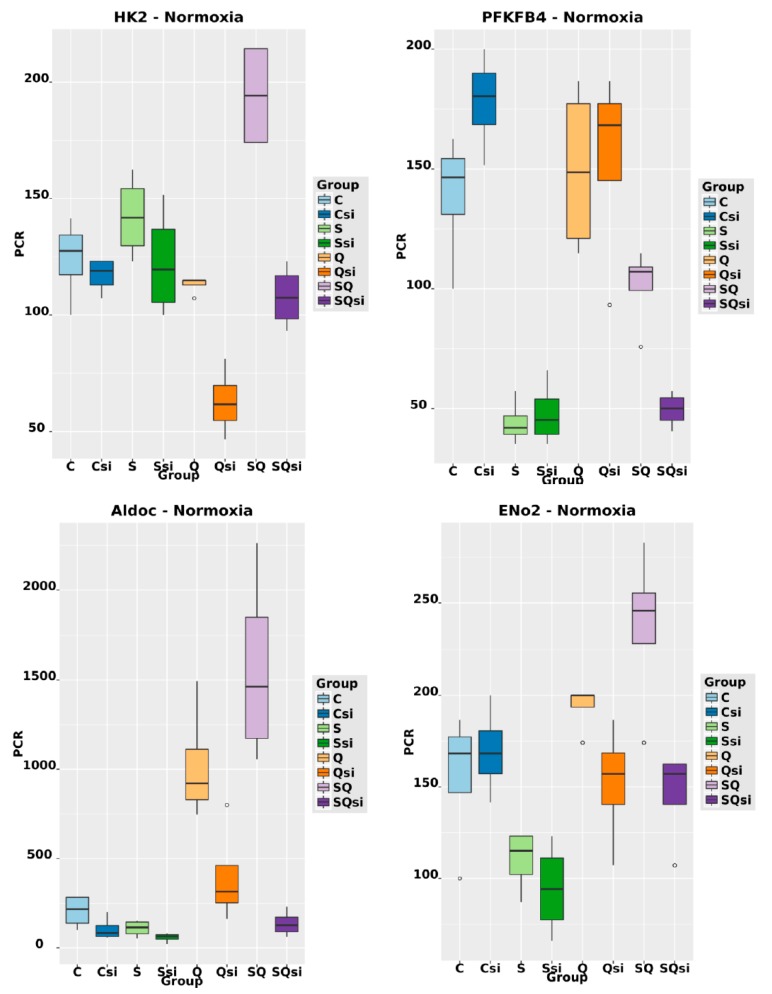
Relative mRNA levels of *HK2, PFKFB4, ALDOC, ENO2, PDK1,* and *LDHA* normalized each to the RPL9 mRNA level, after the application of 5 mM L-glutamine (Q) corresponding to Figure 1 (lanes 5–8), 1% fetal bovine serum (S) (lanes 3, 4, 7, and 8) and/or 5 nM HIF1α-specific siRNA (si) (lanes 2, 4, 6, and 8) or medium without glutamine, serum and siRNA (c -control) lane 1 under normoxic conditions in the MDA-MB-231 cells 24 h after the start of the treatment. The transcription of these genes was significantly regulated by HIF1 under normoxic conditions due to the stabilization of HIF1α via glutamine and fetal bovine serum. (For *p*-values, please refer to Appendix A; see also Appendix A).

**Figure 10 ijms-20-04742-f010:**
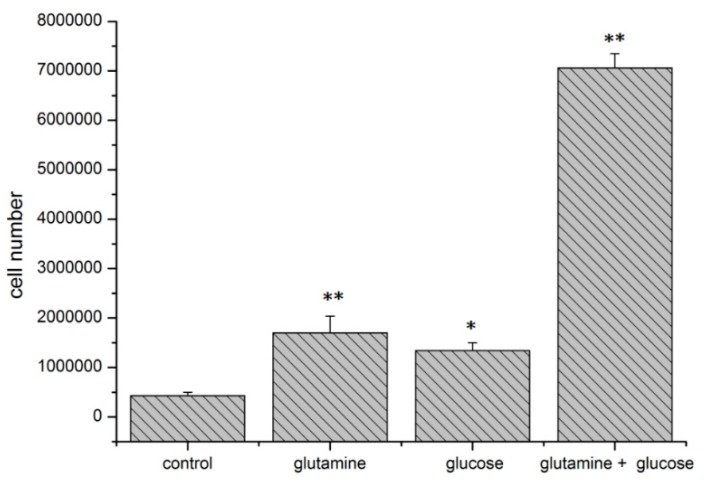
Effect of glutamine and glucose on the cell proliferation of MDA-MB-231 cells. One million cells were cultured in DMEM (with or without glucose and/or glutamine) containing 10% fetal bovine serum (1) without glucose or glutamine, (2) with 5 mM glutamine, (3) with 11.1 mM glucose, and (4) with 5 mM glutamine and 11.1 mM glucose for 72 h. Then, the cells were counted. Compared to the single application of either substance, the use of both glutamine and glucose resulted in rapid cell proliferation. (* -*p*-value < 0.05; ** -*p*-value < 0.01, Student *t*-test as compared to control.

**Figure 11 ijms-20-04742-f011:**
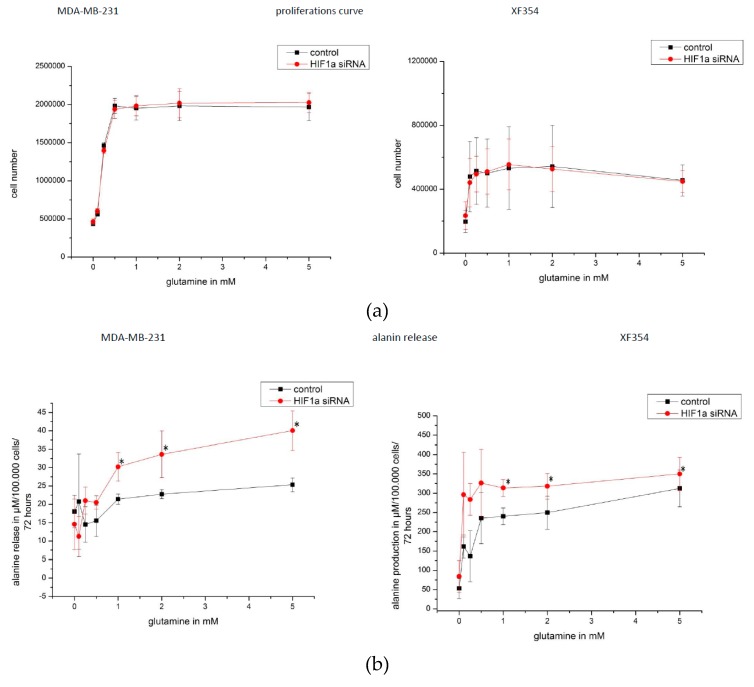
(**a**) Glutamine-mediated effects on proliferation and alanine release in HIF1α-silenced tumor cells. Cell proliferation assay for MDA-MB-231 and XF354 cells were performed with a total of 200,000 cells, which were cultivated for 72 h with different levels of glutamine 0-5 mM in 4 mL medium (RPMI or DMEM each with glucose). The number of cells were counted. MDA-MB-231 cells reached a maximum number at approximately 2,000,000 cells (cultivated in RPMI), and XF354 (cultivated in DMEM) reached a maximum number at approximately 550,000 cells within 72 h of proliferation, respectively. The application of glutamine increased the proliferation of both cells, whereas the treatment with HIF1α-specific siRNA did not. (**b**) The release of alanine, a second waste product of glutamine catabolism, was determined for the MDA-MB-231 and XF 354 cells, which were cultivated for 72 h with different levels of glutamine (0–5 mM in 4 mL of medium). In the MDA-MB-231 treated with lower concentrations of glutamine cells (≤0.5 mM glutamine), the relative alanine level was approximately 20 µM/100,000 cells as compared to the cells that were cultivated with higher levels of glutamine (>1 mM), for which the alanine level was greater. In the presence of 1–5 mM glutamine, a significant increase in the alanine level released by the cells was found for HIF1α-silenced cells compared to the amount of released alanine from the control cells. After glutamine (>0.1 mM) was added to the XF354 cells, the release of alanine was increased such that it was in the range from 100 µM to 300 µM/100,000 cells within 72 h, while the addition of 1–5 mM glutamine led to a significant increase in the alanine level released from cells that were treated with HIF1α-specific siRNA compared to amount of alanine released from the control cells. After 72 h of cultivation in a medium without alanine, the MDA-MB-231 released the maximum alanine concentration at approximately 930 µM, and for XF354 cells, a maximum of 2000 µM alanine was released after 72 h of cultivation. The application of HIF1α-specific siRNA had a significant effect on the amount of alanine released at glutamine levels >1 mM.

**Figure 12 ijms-20-04742-f012:**
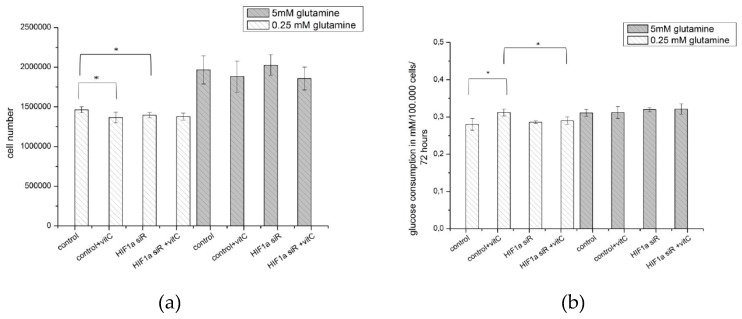
Effect of HIF1α silencing and/or ascorbic acid on the proliferation and metabolic substances under normoxia (**a**,**b**) The results from the analysis of the therapeutic effect of ascorbic acid (vitc) in MDA-MB-231 cells cultivated for 72 h under normoxic conditions with 0.25 mM or 5 mM glutamine and with or without HIF1α-specific siRNA. This experiment was initiated with 200,000 cells. After 72 h of proliferation, the cell number was higher with 5 mM glutamine treatment then it was with 0.25 mM glutamine treatment (see also Figure 11a). At low glutamine levels, the application of ascorbic acid or siRNA against HIF1α resulted in a very limited reduction of the cell number. At a higher level of glutamine application (5 mM), such an effect could not be found. The uptake of glucose from the medium was approximately 0.3 mM glucose /100,000 cells and marginally lower compared to the uptake after ascorbic acid treatment. (**c**) Results from the analysis of the release of lactate from the MDA-MB-231 cells into the medium. The lactate level/100,000 cells within 72 h was marginally increased by the application of ascorbic acid but only at low levels of glutamine. (**d**,**e**) The release of ammonium from the MDA-MB-231 cells into the medium was analyzed. A significant increase in the release of ammonium due to the influence of ascorbic acid, but not HIF1 siRNA, was found at low levels of glutamine (0.25 mM). At high levels of glutamine (5 mM), neither HIF1α-specific siRNA nor ascorbic acid had any effect. The release of alanine was also analyzed for the MDA-MB-231 cells. At lower levels of glutamine (0.25 mM), ascorbic acid significantly reduced the concentration of alanine released, independent of HIF1α silencing, while at higher concentrations of glutamine (5 mM), ascorbic acid induced the opposite effect, with an increased release of alanine into the medium surrounding the cultivated cells. This effect was HIF1α-dependent, which enhanced the concentration of alanine released into the medium. (**f**) Results from the analysis of the ratio of released ammonium to alanine; this ratio was strongly dependent on the presence of ascorbic acid. At low glutamine levels, the application of ascorbic acid caused a HIF1α-independent increase in the ratio, which suggested the release of higher levels of ammonium, but lower levels of alanine. At a higher concentration of glutamine (5 mM), this correlation seems to be inverse of the situation at lower glutamine levels. (* −*p*-value < 0.05; ** −*p*-value < 0.01, Student *t*-test as compared to control.

**Figure 13 ijms-20-04742-f013:**
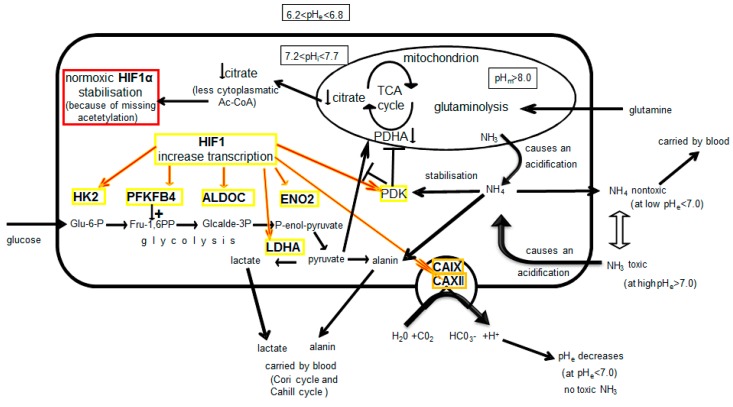
A model for the role of normoxic HIF1 in the regulation of glycolysis and glutaminolysis. The uptake of the amino acid glutamine and the subsequent deamination called glutaminolysis occurs at high rates in tumor cells [13]. The process of glutaminolysis is completed inside the mitochondria, and the mitochondrial pH value (pH_m_) is higher than the pH value in the cytosol. This model demonstrates that the consumption of amino acids such as glutamine reaches a critical point for the pH value-dependent release of ammonia. As the catabolism of glutamine usually occurs in the mitochondrial matrix, the basic intra-mitochondrial pH value leads to the increased concentration of toxic ammonia. Gaseous ammonia, but not ammonium, can diffuse across all membranes, even the impermeable membrane of a mitochondrion, but it will be protonated to ammonium in the intramembranous space, which is populated by protons. The equilibrium between ammonia and ammonium is ph value dependent. The ammonia gas is generated only at a pH value >7.0. Diffuse ammonia from the mitochondria decreases the cytoplasmic pH value. Extracellularly excreted ammonium is able to reenter the cell again as gaseous ammonia when the extracellular pH value is higher than 7.0, which also leads to decreased pH_i_. Ammonia/ammonium is able to activate pyruvate dehydrogenase kinase (PDK) [56,57]. PDK is able to inhibit pyruvate dehydrogenase (PDHA), which is essential for the function of the tricarboxylic acid (TCA) cycle [57]. Such an ammonia-triggered mechanism shut down the TCA cycle, which leads to a reduced level of mitochondrial citrate. Citrate is necessary for the generation of cytoplasmic acetyl-CoA (Ac-CoA), which is required for the modification of proteins through acetylation. The subunit HIF1α is not only modified by hydroxylation but also by acetylation. Low levels of cytoplasmic Ac-CoA inhibit the acetylation of normoxic HIF1α, which results in the stabilization of HIF1α [33]. The accumulation of HIF1α enables the transcriptional activity of HIF1 (composed of two subunits, α and β). This result could explain the observed increase in the mRNA levels of HIF1 target genes (HK2, PFKFB4, ALDOC, ENO2, LDHA, PDK1, CAIX, and CAXII). Thus, it seems that normoxic HIF1 has several mechanisms by which it compensate for the negative effects of the generation of ammonia/ammonium, including (i) stabilizing the energy balance by increasing the rate of glycolysis through the transcriptional activation of glycolysis-associated genes, when ammonia/ammonium is suppressing the activity of the TCA cycle. (ii) Additionally, it stabilizes the intracellular pH value by reducing the extracellular pH value via HIF1-activated CAIX and CAXII, which makes it impossible for ammonia to reenter the cell (at pH_e_ < 7.0). Moreover, the excretion of ammonium requires energy, and the reentry of ammonia into the cell requires additional energy for excretion of ammonium. (iii) Furthermore, the enhancement of glycolysis increases the level of pyruvate, which is able to bind ammonium and form alanine, which can be further regenerated, in a manner that is similar to lactate in the Cori cycle. Indeed, 50% of the surplus nitrogen is secreted as alanine (50% as ammonium) out of cells [8]. In addition, lactate can induce glutamine uptake and metabolism [58] and, then, whole cycle as described. Altogether, the toxic nature of ammonia requires a transcriptional reaction of the genome, which is, in our interpretation, the activation of the transcription factor HIF1. Therefore, normoxic HIF1 functions as a pH sensor, rather than an oxygen/hypoxia sensor, and it is involved in glycolysis and glutaminolysis to maintain the energy and metabolic balance of the cell. (Figure adapted from [13] Figure 5).

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
