# Peer review of "Causes and Consequences of A Glutamine Induced Normoxic HIF1 Activity for the Tumor Metabolism"

_ijms, 2019, doi:10.3390/ijms20194742_

Round 1

Reviewer 1 Report

The manuscript titled “Causes and consequences of a glutamine induced 2 normoxic HIF1 activity for the tumor metabolism” describes the role of glutamine metabolism in HIF1 activation and metabolic outcomes of HIF1 activation in normoxic conditions. The data are interesting and deserve further consideration for publication. However there are some concerns with the data presentation and the paper organization:

1. It is not clear what exactly the authors wish to convey in the first section of Results (2.1. Consequence of an ammonia impulse on the intracellular pH-value) and Fig. 1. The data don’t seem to connect well to the rest of the manuscript or relevant to the major conclusion of the work. The authors are suggested to delete this part from the manuscript.

2. The description of the results should be more explicit. In many places including Abstract, the authors use vague words such as “affected, regulated…” These words don’t tell whether the processes are positively or negatively regulated. The authors are suggested to make changes throughout the text, in particular in Abstract or conclusions.

3. Fig. 5 should include blotting with anti-ACLY, ACSS2 antibodies to show the silencing effects of siRNAs on the corresponding target genes.

4. There are many examples that HIF1 can be upregulated in normoxic conditions by growth factors (EGF), environmental nutrients  (glucose) and activated oncogenes via inhibiting autophagy–mediated degradation of HIF1α (See many research and review articles from Dr. Semenza GL’s group.). The hypoxia-independent HIF1 activity has been well known and reported to influence cancer cell metabolism. It is an overstatement that the data described in the paper show for the first time a function of HIF1 under normoxic conditions by affecting the tumor cell metabolism.

5. Ascorbic acid and vitamin C are not equivalent. When ascorbic acid was used in the experiment (Fig. 6), figure legend should use the compound name instead of Vitamin C.

6. The authors should use names of genes consistently in the text. Examples include CA9/CAIX. CAIX should be used instead of sudden change to CA9 in Fig. 7. In addition, Fig. 7 should be expanded to include analysis of the effects of acetylation agents on CAIX mRNA.

7. In general, there is English issue in the Abstract and Text. Certain sentences in Abstract are difficult to understand, e.g. “However, targeted metabolome analyses, HIF1 activity does neither affect the consumption of glucose nor the release of ammonium or lactate, but the release of 56 the amino acid alanine”. Do the authors mean “through targeted metabolome analyses…? The whole paper also looks very loose. The Discussion section is too long. In Results section, some figures could be combined into panels of a single figure. The current way of data presentation is based on individual experiments instead of scientific questions to be addressed or idea points to be made.

Author Response

Dear reviewer,

At first we would like to thank all of you for your effort with our demanding and really long manuscript and for all your very helpful questions and comments! We have tried to answer and to respond to all of them. In addition, we performed requested experimental work, shortened the text of the manuscript and involved native speakers for proofreading.

Please, let us to respond in a point-by-point way:

Reviewer 1

Comments and Suggestions for Authors

The manuscript titled “Causes and consequences of a glutamine induced 2 normoxic HIF1 activity for the tumor metabolism” describes the role of glutamine metabolism in HIF1 activation and metabolic outcomes of HIF1 activation in normoxic conditions. The data are interesting and deserve further consideration for publication. However there are some concerns with the data presentation and the paper organization:

It is not clear what exactly the authors wish to convey in the first section of Results (2.1. Consequence of an ammonia impulse on the intracellular pH-value) and Fig. 1. The data don’t seem to connect well to the rest of the manuscript or relevant to the major conclusion of the work. The authors are suggested to delete this part from the manuscript.

Answer: We followed the suggestion of the reviewer and removed these sections from the chapter result. However, since the effect of ammonia / ammonia is important for the purposes of this manuscript, the results of this section have been included in the supplementary figure (S1) to allow the readers to view these basic experiments.

 The description of the results should be more explicit. In many places including Abstract, the authors use vague words such as “affected, regulated…” These words don’t tell whether the processes are positively or negatively regulated. The authors are suggested to make changes throughout the text, in particular in Abstract or conclusions.

Answer: Many thanks for this helpful comment. Yes, we have revised the manuscript and integrated the direction of changes (up/down) or its interpretation (negative/positive) throughout the manuscript.

Fig. 5 should include blotting with anti-ACLY, ACSS2 antibodies to show the silencing effects of siRNAs on the corresponding target genes.

Answer:  Both siRNAs are commercial siRNAs. We established an ACSS2 antibody (abcamAb66038) that could detect the ACSS2 knockdown. However, the establishment of 3 different ACLY antibodies was not successful because these antibodies detected either other proteins (wrong protein size) or did not work at all. Therefore we successfully established an ACLY  Antibody against phosphor ser455 (cell signaling 4331) because the cell signaling 4332 against ACL do not work in our hands.We added the data for these proteins from the same blot in Figure 4

There are many examples that HIF1 can be upregulated in normoxic conditions by growth factors (EGF), environmental nutrients  (glucose) and activated oncogenes via inhibiting autophagy–mediated degradation of HIF1α (See many research and review articles from Dr. Semenza GL’s group.). The hypoxia-independent HIF1 activity has been well known and reported to influence cancer cell metabolism. It is an overstatement that the data described in the paper show for the first time a function of HIF1 under normoxic conditions by affecting the tumor cell metabolism.

Answer:  We agree with the reviewer. We have focused our statement to: “This study comprehensively investigated, for the first time, how normoxic HIF1 is stabilized, and it analyzed the possible function of normoxic HIF1 in the transcriptome and metabolic processes of tumor cells in a breast cancer cell model.  ” In addition, we have added a recent article on therapeutic approaches to HIF1 to discuss these important facts as well. (Semenza 2019, Annu Rev Pharmacol Toxicol. 2019 Jan 6;59:379-403.)

Ascorbic acid and vitamin C are not equivalent. When ascorbic acid was used in the experiment (Fig. 6), figure legend should use the compound name instead of Vitamin C.

Answer:  We used an L-ascorbic acid, which was sold as Vitamin c. But we agree and changed throughout the manuscript the term vitamin c into ascorbic acid. Many thanks for that important note.

The authors should use names of genes consistently in the text. Examples include CA9/CAIX. CAIX should be used instead of sudden change to CA9 in Fig. 7. In addition, Fig. 7 should be expanded to include analysis of the effects of acetylation agents on CAIX mRNA.

Answer:  We changed the term for mRNA, therefore we wrote CA9, for the gene: CAIX und for the protein: CAIX. Many thanks for that note.

In general, there is English issue in the Abstract and Text. Certain sentences in Abstract are difficult to understand, e.g. “However, targeted metabolome analyses, HIF1 activity does neither affect the consumption of glucose nor the release of ammonium or lactate, but the release of 56 the amino acid alanine”. Do the authors mean “through targeted metabolome analyses…? The whole paper also looks very loose. The Discussion section is too long. In Results section, some figures could be combined into panels of a single figure. The current way of data presentation is based on individual experiments instead of scientific questions to be addressed or idea points to be made.

Answer: The manuscript was reduced to the most important facts. Proofreading of the manuscript was performed by native speakers (AJE).

Thank you for all your helpful comments.

Submission Date

31 May 2019

Date of this review

14 Jun 2019 18:19:40

Reviewer 2 Report

The datasets about the role of hypoxia, glutamine and other reagent effect on cancer cell line is reasonable. However, sentences are very hard to read throughout the manuscript, with lots of grammatical errors I gave up to list up. I had to stop and think about multiple possibilities of interpretation many times. These flaws made hard to judge if the conclusion they draw and their interpretation is reasonable or not. Thorough re-writing and brush-up are needed.

L63-L75 Could be written better. What about describing Warburg effect first, glutamine next, and then combine the two leading to the purpose of this study? Relationship between HIF1 stabilization and glutamine should be also described here to introduce readers to the main topic of this paper.

Figures 3-6 should be accompanied with quantification of Western blots signal intensities, how many times the experiments are repeated, and their statistical analysis.

L103-L118 The purpose of this experiment is not clear. What is expected by NH4Cl (mixed with CL) prepulse and what figure 1 is expected to evaluate? How it relates to HIF (L117)? Brief description about how intracellular pH was measured is necessary here.

L136 "some scientists are of opinion that" specifically refer to the scientist

L136-140 Which is utilized by cancer cells, ammonia or ammonium ion? and modify other description accordingly

L144-146 Describe the cause and consequence separately and clearly. Description of the relationship between glutamine and ammonia is necessary.

L148 Where "these data" are shown?

L153 Abbreviation and explanation of CAIX is necessary.

L178 Does it mean either hypoxic condition or glutamine cause HIF1a stabilization?

L184 This sentence is not complete.

L185-L190 Analysis of HIF1a acetylation is preferred, because acetylsalicylic acid and ibuprophen are COX inhibitors and COX inhibition may affect HIF1a expression. Also, "expression" indicate transcriptional or translational effect on the amount of the protein, and should not use if you meant to study stabilization.

L203-212 Similarly, acetyl-CoA is a starting substrate of Krebs cycle and oxidative phosphorylation, so its depletion will also indirectly affect HIF1a availability.

L219 "independent of glutamine" should be confirmed by glutamine application and comparison with acetylation effects.

L229 "treatment with vitamin C had only a very weak effect" although ascorbic acid did not completely deplete HIF1a, but I see significant reduction.

L234 Need explanation of CA9. Also consistently use Ca9 or CA9. Describe how the CA9 mRNA was quantified?

L289 Graph X-axis labels and legends are redundant. Mark for pairs with significant difference

Discussion should be made more concise. Instead of reviewing all aspects of cancer cell metabolism, just briefly mention the most important and relevant facts and discuss what are the impacts of this study. It will be better to keep the length of discussion shorter than the results section.

Author Response

Dear Mr. Tang and dear reviewers,

At first we would like to thank all of you for your effort with our demanding and really long manuscript and for all your very helpful questions and comments! We have tried to answer and to respond to all of them. In addition, we performed requested experimental work, shortened the text of the manuscript and involved native speakers for proofreading.

Please, let us to respond in a point-by-point way:

Reviewer 2

Comments and Suggestions for Authors

The datasets about the role of hypoxia, glutamine and other reagent effect on cancer cell line is reasonable. However, sentences are very hard to read throughout the manuscript, with lots of grammatical errors I gave up to list up. I had to stop and think about multiple possibilities of interpretation many times. These flaws made hard to judge if the conclusion they draw and their interpretation is reasonable or not. Thorough re-writing and brush-up are needed.

L63-L75 Could be written better. What about describing Warburg effect first, glutamine next, and then combine the two leading to the purpose of this study? Relationship between HIF1 stabilization and glutamine should be also described here to introduce readers to the main topic of this paper.

Answer: L63-L75 were revised…We add after the term Aerobic glycolysis (L63) the explanation (the so called Warburg effect)  because aerobic glycolysis is a synonym for the Warburg effect. In that way we already described the Warburg effect first, as requested by the reviewer 2, but do not used the term Warburg effect so far and only the term aerobic glycolysis, which was now completed by using both term. Therefore, we described the Warburg effect first ( L63-L69) and then as proposed the importance of the metabolism of glutamine (glutaminolysis) L70-L76). Then as proposed, the combination of both pathways were described (L76-L79).  

After the introduction of HIF1, which is necessary in our opinion, L80-L89, the expected association of HIF1 activity and glycolysis /glutaminolysis were described L89-L100.

Because of the suggestion of the reviewer 2, we added the following sentence “We already described an accumulation and activity of normoxic HIF1 because of the catabolisation of the amino acid glutamine and by the release of the waste product of glutaminolysis ammonia [13,31,32].“ which should describe the main topic of the results of our manuscript

Figures 3-6 should be accompanied with quantification of Western blots signal intensities, how many times the experiments are repeated, and their statistical analysis.

Answer:  We did the experiments of Fig 3-6 at least 3 times and Western blot intensities were added ( Please see supplemental Figures).

L103-L118 The purpose of this experiment is not clear. What is expected by NH4Cl (mixed with CL) prepulse and what figure 1 is expected to evaluate? How it relates to HIF (L117)? Brief description about how intracellular pH was measured is necessary here.

Answer: The results of L103-L118 of that experiments were translocated to supplemental Figures 1, because of the suggestion of three reviewers we decided to remove that data from the chapter results.

L136 "some scientists are of opinion that" specifically refer to the scientist

Answer: This was described in the chapter discussion by using the arguments of Spinelli et al.. Spinelli et al. suggested that the gas ammonia is metabolic used by tumor cells. The results of L136 were deleted from the results chapter as well as from the chapter discussion. However, I do not believe that these data (by Spinelli et al. ) are consistent since ammonia is still a poisonous gas and cannot be controlled or used. However, we removed this confrontation with this data from Spinelli et al., from this manuscript.

L136-140 Which is utilized by cancer cells, ammonia or ammonium ion? and modify other description accordingly

Answer: Please see also Spinelli et al…, these authors described that the gas ammonia is used by tumor cells, which is in our opinion not a necessary explanation of the results of Spinelli et al. and can be refuted by our own experiments ( supplemental figure 2). Whether Spinelli et al additionally use glutamine to study the influence of ammonia or not is hardly recognizable.  Spinelli et al. wrote in the Supplemental part :Metabolic Assays:”Following this incubation, media was changed to a non-buffered, serum-free Seahorse Media (Seahorse Bioscience, Catalog #102353) supplemented with 5 mM glucose, 2 mM L-glutamine, 1 mM sodium pyruvate, and the appropriate ammonium concentration.” For me, it seems that they used extra glutamine to study the effect of ammonium/ammonia, which would not be a convincing experimental approach. However, we removed this confrontation with this data from Spinelli et al., from this manuscript.

L144-146 Describe the cause and consequence separately and clearly. Description of the relationship between glutamine and ammonia is necessary.

Answer: We transferred this chapter into the chapter supplemental Figures. The relationship of glutamine and especially the relationship of ammonia and ammonium requires a deep understanding of chemical and biochemical knowledge and requires pages to be explained clearly and in detail. Therefore, we gave just some general/introducing informations. For a deep understanding a study of chemical and biochemical specialist books is necessary, because not all general informations which are written in thoses books can be explain in such a short manuscript. Moreover, all reviewers suggested to reduce the chapter discussion, therefore, we reduced the general information’s regarding fundamental chemical and biochemical informations to a minimum.

L148 Where "these data" are shown?

Answer:The data were transferred to the chapter supplemental data.

L153 Abbreviation and explanation of CAIX is necessary.

Answer: We thank reviewer 2 for mentioning that explanation for CAIX is necessary. We have included in the result chapter: Carbonic anhydrase (CA9) is a target of the transcription factor HIF1a [16]. (L245) Moreover, in L163 we already wrote “the HIF1 target carbonic anhydrase (CAIX), a regulator of extracellular pH value, was only detectable in the presence of HIF1α”. We added Carbonic anhydrase before the abbreviation CAIX.

L178 Does it mean either hypoxic condition or glutamine cause HIF1a stabilization?

Answer: Yes, either hypoxia or under normoxic conditions; glutamine causes HIF1a stabilization.

L184 This sentence is not complete.

Answer:  Proofreading of the text was performed by a native speaker ( AJE).

We revised the sentence to:” In addition, the stability of HIF1α is also decreased by the acetylation of the α-subunit of HIF1 [33].”

L185-L190 Analysis of HIF1a acetylation is preferred, because acetylsalicylic acid and ibuprophen are COX inhibitors and COX inhibition may affect HIF1a expression. Also, "expression" indicate transcriptional or translational effect on the amount of the protein, and should not use if you meant to study stabilization.

Answer: Yes, the reviewer has of course right; acetylsalicylic acid and ibuprophen are COX inhibitors. For acetylsalicylic acid, we know that it covalently modifies the COX-2 enzyme by acetylating a serine amino acid residue (Lim et al. Arch Pharm Res. 2018).

Tantham et al .2017, however, described an unspecific impact of aspirin on the acetylation of cellular proteins, what helps us to interpret our data. Therefore, we added the following sentence into the chapter discussion:” These results can be explained by the fact that acetylsalicylic acid has the capacity to acetylate the amino group of the lysine side chains in cellular and extracellular proteins [51], and it seems that HIF1α Lys532 is also affected (Figure 3).

L203-212 Similarly, acetyl-CoA is a starting substrate of Krebs cycle and oxidative phosphorylation, so its depletion will also indirectly affect HIF1a availability.

Answer: We agree with the reviewer. But we wanted to concentrate on the level of cytosolic acetyl-CoA that is essential for HIF1a acetylation as described by Jeong et al. 2002. The mitochondrial level of acetyl-CoA, which is indeed important for Krebs cycle and oxidative phosphorylation is not investigated by your experimental design.

L219 "independent of glutamine" should be confirmed by glutamine application and comparison with acetylation effects.

Answer: We agree with the reviewer and we have changed “glutamine-independent/independent of glutamine” to “without glutamine” .

L229 "treatment with vitamin C had only a very weak effect" although ascorbic acid did not completely deplete HIF1a, but I see significant reduction.

Answer: In this case, we compared the effect of ascorbic acid between normoxia and hypoxia. The effect under hypoxia was weaker than that under normoxia. But we corrected this sentence: “Under normoxia, ascorbic acid treatment completely reversed (p=0.01) the glutamine-induced HIF1α protein accumulation in a manner similar to that of HIF1α siRNA silencing (p=0.03) (Figure 6).”

L234 Need explanation of CA9. Also consistently use Ca9 or CA9. Describe how the CA9 mRNA was quantified?

Answer: The term Ca9 was a typing error, thank you for that note. We changed the term for mRNA we wrote CA9, for the gene CAIX und for the protein CAIX.

The CA9-mRNA was measured using Taq-Man Probes (Invitrogene) in duplex analysed together with the reference gene RPL9 and as we described in M&M, for all mRNA the same procedure was used. PCR was performed using a Rotor-Gene RG-6000 (LTF, Wasserburg, Germany) duplex (gene of interest (FAM label probe reference (RPL9) VIC labelled probe). Each gene was analyzed in parallel together with the reference gene RPL9 for normalization using for all samples (Applied Biosystems, Foster City , USA). “The quantification of the transcript amounts was performed by ΔΔCt method.” We added this sentence to M&M.”. Thanks for that important note.

L289 Graph X-axis labels and legends are redundant. Mark for pairs with significant difference

Answer:  We described all significant differences with the associated p value in Suppl. Tab. 1 and Suppl. Tabl.2.

Discussion should be made more concise. Instead of reviewing all aspects of cancer cell metabolism, just briefly mention the most important and relevant facts and discuss what are the impacts of this study. It will be better to keep the length of discussion shorter than the results section.

Answer: Proofreading of the manuscript was performed by native speakers (AJE) and it was significantly reduced.

Thank you for all your helpful comments.

Submission Date

31 May 2019

Date of this review

10 Jun 2019 14:19:08 

Reviewer 3 Report

In their manuscript, Kappler et al. are reporting a study on the effect of glutamine on the HIF1 activity under normoxic conditions in cancer cells. The topic is very interesting as it deals with metabolic changes, including Warburg effect, occurring in cancer cells.

Here are my comments:

-Data shown in figure 1 illustrate the intracellular pH changes elicited by a 5mM pulse of ammonium chloride. There is clearly a run-down of the fluorescence signal which could be attributed to a constitutive acidification of the cells throughout the experiment. In addition, after ammonium chloride exposure, there is an acidification of the cytoplasm. One can ask whether this basal pH instability observed for all measurements. In addition, have the authors tried to monitor intracellular pH longer than 1300seconds to examine whether a pH recovery is observed? It is hard to tell if changes in pH are significant.

-Under normoxic conditions and in the presence of acetylsalicylic acid or ibuprofen, there is a clear blockade of the glutamine-induced HIF1alpha expression. Could this effect involve an antioxidant effect in addition an ‘acetylating’ effect of both acetylsalicylic acid and ibuprofen?

-With regards to experimental data, statistical analysis is not shown for all data presentations. What is the reason for that?

-In my opinion, the Discussion section needs to be shrunk and more focused on data presently shown.

-The manuscript needs to be checked for English language and extensive reading is required to remove grammatical mistakes.

Author Response

Dear Mr. Tang and dear reviewers,

At first we would like to thank all of you for your effort with our demanding and really long manuscript and for all your very helpful questions and comments! We have tried to answer and to respond to all of them. In addition, we performed requested experimental work, shortened the text of the manuscript and involved native speakers for proofreading.

Please, let us to respond in a point-by-point way:

Reviewer 3

Comments and Suggestions for Authors

In their manuscript, Kappler et al. are reporting a study on the effect of glutamine on the HIF1 activity under normoxic conditions in cancer cells. The topic is very interesting as it deals with metabolic changes, including Warburg effect, occurring in cancer cells.

Here are my comments:

-Data shown in figure 1 illustrate the intracellular pH changes elicited by a 5mM pulse of ammonium chloride. There is clearly a run-down of the fluorescence signal which could be attributed to a constitutive acidification of the cells throughout the experiment. In addition, after ammonium chloride exposure, there is an acidification of the cytoplasm. One can ask whether this basal pH instability observed for all measurements. In addition, have the authors tried to monitor intracellular pH longer than 1300seconds to examine whether a pH recovery is observed? It is hard to tell if changes in pH are significant.

Answer: We followed the suggestion of the Reviewer 1&2 and removed these sections from the chapter results. However, since the effect of ammonia / ammonia is important for the purposes of this manuscript, the results of this section have been included in the supplementary figure 1 to allow the readers to view these basic experiments.The experiment was repeated three times in independent experiments each with always >10 cells (please, see also supplemental Figure 1). The calculation demonstrated for all experiments the same results, which was in the end significant (at using student t-Test). In line with our findings, other authors described similar results Chiche, J.; Brahimi-Horn, M.C.; Pouysségur, J. J. Cell. Mol. Med. 2010 and Swietach, P.; Hulikova, A.; Vaughan-Jones, R.D.; Harris, Oncogene 2010 (please, refer to the discussion chapter and the list of references).The final step in these experiments was to permeabilize the cell membrane using a specific calibration solution with a specific pH value. This was necessary to calibrate the intracellular pH measurement. Therefore, the measurement always stops at this time point. However, we have observed different cell lines for more than 24 hours with even 10 mM ammonium chloride in other experiments. We do not find toxic effect, since the ion ammonium is not toxic.

-Under normoxic conditions and in the presence of acetylsalicylic acid or ibuprofen, there is a clear blockade of the glutamine-induced HIF1alpha expression. Could this effect involve an antioxidant effect in addition an ‘acetylating’ effect of both acetylsalicylic acid and ibuprofen?

Answer: Yes, for aspirin also an oxidant effect is known. Recently, Handa et al. (Free Radic Res. 2018) reported an acetyl salicylic acid-induced injury of the small intestine cells that was reasoned by increased reactive oxygen species production. This effect could be reversed by antioxidant treatment. However, we wanted to concentrate on the “acetylating effect” of acetylsalicylic acid and ibuprofen.

Furthermore, it was described, that aspirin has the capacity to acetylate the ἑ-amino-group of lysine side-chains in cellular and extracellular proteins and this aspirin-mediated lysine acetylation could explain some of its as-yet unexplained drug actions or side-effects. (Tatham et al 2017.Doi: 10.1074/mcp.O116.065219). As we already mentioned in the manuscript,   Jeong et al. first described an ARD-1 (NAA10 N(alpha)-acetyltransferase 10, NatA catalytic subunit (NM_003491)) mediated acetylation of Lys532 in HIF1α, which causes the normoxic degradation of HIF-1α. Therefore, it is more likely that indeed the acetylation of Lys532 in HIF1α could be responsible for the degradation and the absence of HIF1α protein in our aspirin treated samples. We added the paper   (Tatham et al 2017) to our reference list.

-With regards to experimental data, statistical analysis is not shown for all data presentations. What is the reason for that?

Answer: We have performed statistical analysis for the RNA-Sequencing data and for the “Therapeutical effects” of ascorbic acid (Fig. 12). (Please, refer also to M&M chapter: 4.9. Bioinformatics approach and Statistical methods. The statistical analysis of the western blot data was added as supplemental figures.

-In my opinion, the Discussion section needs to be shrunk and more focused on data presently shown.

Answer: We agree. Therefore, the whole manuscript was significantly reduced and especially the discussion chapter.

-The manuscript needs to be checked for English language and extensive reading is required to remove grammatical mistakes.

Answer: We agree. Therefore, proofreading of the manuscript was performed by native speakers ( AJE) and it was significantly reduced.

Thank you for all your helpful comments.

Submission Date

31 May 2019

Date of this review

12 Jun 2019 15:40:43

Reviewer 4 Report

The study entitled, “Causes and consequences of a glutamine induced normoxic HIF1activity for the tumor metabolism” by Kappler et al. investigates the causes of the accumulation of HIF1 under normoxic conditions and the effects of HIF1 on tumor cell proliferation and tumor metabolism. The major concern is the significance of the study and there are several conclusions without supporting data. This study did not add substantially to the authors’ previous study “Normoxic accumulation of HIF1a is associated with glutaminolysis”. Specific comments are outlined below. 

Major Comments:

1. Figure      1 only indicates that “This experiment demonstrated the effect of      extrinsic ammonia on the intracellular pH value” but the authors made the      conclusion “could explain why HIF1 is also stabilized and activated even      in a normoxic environment, just to stabilize the pH value over a longer      period of time through its transcriptional function” without any      supporting data.

2. Question      whether the dose of 5mM NH4Cl is the physiological dose for cellular      media.

3. The      authors conclude that “the expression of HIF1 and CAIX is not      hypoxia-specific” which contradicts the data presented in Figure 2: high      expression of HIF1 under hypoxia. The bar graphs representing the relative      HIF1a protein level/b-actin also contradicts the western blot image which      shows no expression of HIF in lane 1, 2, 3, 4, 6, 8, 2*, 4*, 6*, and 8*.

4. The      authors conclude in Section 2.4 that “Under normoxia, vitamin C treatment      completely reversed the glutamine-induced HIF1a protein accumulation,      similar to the effects of HIF1a siRNA silencing,” when HIF1a siRNA did not      “completely reverse the glutamine-induced HIF1a protein accumulation” as      shown in lane 2, 3, 4 in Figure 6.

5. The authors      should consider adding in Section 2.4 why CA9 nRNA level is measured, and      how it is related to the level and the activity of normoxic HIF1.

6. The      authors state in Section 3.5 that “Ascorbic acid leads to a small, but      nevertheless significantly reduced cell number in the investigated tumor      cell line, but only at a low glutamine level of 0.25 mM (physiological      level of glutamine) (Figure 12a).” “Small” and “significantly reduced”      seem to contradict each other. The authors should consider providing      evidence that the cell number is “significantly reduced.”

Minor Comments:

1. The authors should consider using two different colors for normoxia and hypoxia in Figure 2 and Figure 7.  

2. The authors should consider being consistent whether to use abbreviation HIF1 or hypoxia-inducible factor 1 in Section 3.1. E.g. “The hypoxia-inducible factor 1 can be normoxic activated due to ammonia concentrations >50-100 μM [13].” “It seems that the effects of ammonia / ammonium on HIF1 we have found [13,32], have long since been confirmed and some researchers are already trying to derive applications from this [42,43].”

3. The authors should consider specifying what the abbreviations mean in Figure S5 and Figure S6.

4. The authors should consider making Figure 2, Figure 11, and Figure 12 more clear.

5. Minor formatting and spelling issues but does not affect the readability.

Author Response

Dear Mr. Tang and dear reviewers,

At first we would like to thank all of you for your effort with our demanding and really long manuscript and for all your very helpful questions and comments! We have tried to answer and to respond to all of them. In addition, we performed requested experimental work, shortened the text of the manuscript and involved native speakers for proofreading.

Please, let us to respond in a point-by-point way:

Reviewer 4

Comments and Suggestions for Authors

The study entitled, “Causes and consequences of a glutamine induced normoxic HIF1activity for the tumor metabolism” by Kappler et al. investigates the causes of the accumulation of HIF1 under normoxic conditions and the effects of HIF1 on tumor cell proliferation and tumor metabolism. The major concern is the significance of the study and there are several conclusions without supporting data. This study did not add substantially to the authors’ previous study “Normoxic accumulation of HIF1a is associated with glutaminolysis”. Specific comments are outlined below. 

Major Comments:

Figure      1 only indicates that “This experiment demonstrated the effect of      extrinsic ammonia on the intracellular pH value” but the authors made the      conclusion “could explain why HIF1 is also stabilized and activated even      in a normoxic environment, just to stabilize the pH value over a longer      period of time through its transcriptional function” without any      supporting data.

Answer: We agree with the reviewer, Fig. 1 only shows effect of extrinsic ammonium on the intracellular pH value. The conclusion for the stabilization of HIF1 under normoxia was based on results in reference 13 (Kappler et al. Clinical oral investigations 2017). Since this part was in its present form somewhat confusing and reviewer 1,2,3 asked to remove it to the supplemental part of the manuscript. In this way the manuscript was shortened in the result chapter and even for a larger part in the discussion chapter.We investigated the extrinsic ammonia instead of the intrinsic ammonia, because intrinsic ammonia is released probably by the mitochondria. It is difficult to find a valid experimental design to use the intracellular mitochondria to release specific amount of ammonia in a specific time. In this way, such an approach remains for future experiments.

Question      whether the dose of 5mM NH4Cl is the physiological dose for cellular      media.

Answer: To generate levels of ammonia gas, which can enter the cells, it is necessary to use high amounts of NH4Cl, because the ph value around the cells cannot be increased as high. Spinelle et al., already used 50 mM NH4Cl in their experiments. So it is common to used such high levels of NH4Cl. However ,in the renal system levels of 5mM NH4Cl are possible.

The      authors conclude that “the expression of HIF1 and CAIX is not      hypoxia-specific” which contradicts the data presented in Figure 2: high      expression of HIF1 under hypoxia. The bar graphs representing the relative      HIF1a protein level/b-actin also contradicts the western blot image which      shows no expression of HIF in lane 1, 2, 3, 4, 6, 8, 2*, 4*, 6*, and 8*.

Answer: We could show not only under hypoxia but also at 1% serum+glutamine a strong protein expression of HIF1a and CA9. Therefore, we concluded that the expression of HIF1 and CAIX is not hypoxia-specific and, an oxygen independent mechanism must be responsible for the HIF1 stabilization under normoxia.

The scale on the y-ordinate is logarithmic.

Answer: We agree with the reviewer that with the naked eye no HIF1a protein expression can be seen in lane 1, 2, 3, 4, 6, 8, 2*, 4*, 6*, and 8*. But with the FUSION-FX7 SPECTRA Chemiluminescence-Fluorescence-Imaging System (Vilber Lourmat, Eberhardzell, Germany) still weak signals could be measured and are reported.

The      authors conclude in Section 2.4 that “Under normoxia, vitamin C treatment      completely reversed the glutamine-induced HIF1a protein accumulation,      similar to the effects of HIF1a siRNA silencing,” when HIF1a siRNA did not      “completely reverse the glutamine-induced HIF1a protein accumulation” as      shown in lane 2, 3, 4 in Figure 6.

Answer: Yes, we wrote: “Under normoxia, ascorbic acid treatment completely reversed (p=0.01) the glutamine-induced HIF1α protein accumulation in a manner similar to that of HIF1α siRNA silencing (p=0.03) (Figure 6). We agree in lane 3 (siRNA treatment against HIF1), there is still HIF1 protein detectable but this was very apparently reduced compared to the not siRNA treated cells. Vitamin C reversed completely the glutamine-induced HIF1a protein accumulation, i.e., no HIF1a protein can be seen in the Western blot. We wrote that this is similar [but not that it is the same] to the siRNA effect. We still think that this is correct. But any other suggestion/description of this result is appreciated.

Furthermore, in the same experiment the CA9-mRNA level was analyzed. This indirectly represents the concentration of active HIF1 and demonstrated that there is always some active HIF1 protein in very low levels in the cells.  5. The authors      should consider adding in Section 2.4 why CA9 nRNA level is measured, and      how it is related to the level and the activity of normoxic HIF1.

Answer: We thank reviewer 4 for this comment since carbonic anhydrase 9 (CA9) was not introduced. CA9 is a target of the transcription factor HIF1a (reference 16: Semenza, Nat Rev Cancer 2003). Therefore, mRNA level of CA9 is regulated by HIF1a, both when HIF1a is induced by hypoxia (lane 5) and by glutamine under normoxic conditions (lane 1). But at adding vitamin C the amount of HIF1a is reduced both under hypoxia (lane 6) and under normoxia (lane 2) (please, consider again that the Y-ordinate has a logarithmic scale).

The mRNA as well as the protein level of the CAIX gene is strongly dependent on the level of active HIF1 protein since HIF1 is the most important transcription factor for CAIX (or an important enhancing factor).

The      authors state in Section 3.5 that “Ascorbic acid leads to a small, but      nevertheless significantly reduced cell number in the investigated tumor      cell line, but only at a low glutamine level of 0.25 mM (physiological      level of glutamine) (Figure 12a).” “Small” and “significantly reduced”      seem to contradict each other. The authors should consider providing      evidence that the cell number is “significantly reduced.”

Answer: We performed Unpaired Student’s t-test to test statistical significance (Please see supplemental Figures). Based on this test, we detected small but significant differences between the control and the vitamin C treated cells (Fig. 12a,b and c). Small and significant is not a contradiction.

Minor Comments:

The authors should consider using two different colors for normoxia and hypoxia in Figure 2 and Figure 7.  

Answer: We agree to this possibility. But we think that the reader can already distinguish between normoxia and hypoxia since it is given as head-line of the experiments.

The authors should consider being consistent whether to use abbreviation HIF1 or hypoxia-inducible factor 1 in Section 3.1. E.g. “The hypoxia-inducible factor 1 can be normoxic activated due to ammonia concentrations >50-100 μM [13].” “It seems that the effects of ammonia / ammonium on HIF1 we have found [13,32], have long since been confirmed and some researchers are already trying to derive applications from this [42,43].”

Answer: We corrected this throughout the manuscript, thank you for this note.

The authors should consider specifying what the abbreviations mean in Figure S5 and Figure S6.

Answer: We described this for Supplemental figure S13, because the number of all figures have changed now.  H means –hypoxia , N- normoxia, S-serum application and si – application of HIF1 specific siRNAs

The authors should consider making Figure 2, Figure 11, and Figure 12 more clear.

Answer: We have shortened figure legends of Fig. 2, 11 and 12 to make their content and conclusion clearer.

Minor formatting and spelling issues but does not affect the readability.

Answer: Proofreading of the manuscript was performed by native speakers ( AJE) and it was significantly reduced.

Many thanks for this and the other positive comments.

Submission Date

31 May 2019

Date of this review

15 Jun 2019 02:38:18

Altogether, we would like to thank Mr. Tang and all four reviewers for their great effort again.

Sincerely yours,

Matthias Kappler Ph.D.

Round 2

Reviewer 1 Report

The authors addressed several of my concerns and improved the writing.

Reviewer 2 Report

I see the authors has significantly improved this manuscript taking reviewers suggestions, and could be accepted.